# Mutational scanning pinpoints distinct binding sites of key ATGL regulators in lipolysis

Johanna M. Kohlmayr[1], Gernot F. Grabner [2,3], Anna Nusser[1], Anna Höll [1], Verina Manojlović[1], Bettina Halwachs [1,4], Sarah Masser [1,5], Evelyne Jany-Luig [1], Hanna Engelke [1,4], Robert Zimmermann[2,4,5] & Ulrich Stelzl [1,4,5] ✉

ATGL is a key enzyme in intracellular lipolysis and plays an important role in metabolic and cardiovascular diseases. ATGL is tightly regulated by a known set of protein-protein interaction partners with activating or inhibiting functions in the control of lipolysis. Here, we use deep mutational protein interaction perturbation scanning and generate comprehensive profiles of single amino acid variants that affect the interactions of ATGL with its regulatory partners: CGI-58, G0S2, PLIN1, PLIN5 and CIDEC. Twenty-three ATGL amino acid variants yield a specific interaction perturbation pattern when validated in co-immunoprecipitation experiments in mammalian cells. We identify and characterize eleven highly selective ATGL switch mutations which affect the interaction of one of the five partners without affecting the others. Switch mutations thus provide distinct interaction determinants for ATGL's key regulatory proteins at an amino acid resolution. When we test triglyceride hydrolase activity in vitro and lipolysis in cells, the activity patterns of the ATGL switch variants trace to their protein interaction profile. In the context of structural data, the integration of variant binding and activity profiles provides insights into the regulation of lipolysis and the impact of mutations in human disease.

Lipid droplets (LDs) are the main cellular storage organelles for triacylglycerol (TAG). The dysregulation of the TAG metabolism is associated with human metabolic pathologies, including obesity, cardiac pathophysiology, non-alcoholic fatty liver disease, lipodystrophy- and neutral lipid storage disease (NLSD)[1,2]. The highly conserved process of intracellular lipolysis mediates the hydrolysis of TAG stored in LDs. Adipose triglyceride lipase (ATGL; gene name: *PNPLA2*) is the key enzyme in lipolysis and catalyzes the first step of TAG breakdown by cleaving off the first fatty acid from the TAGs resulting in a diacylglycerol (DAG) and free fatty acids[3]. The catalytic activity of ATGL is

important for the regulation of whole-body energy homeostasis and lack of functional ATGL leads to severe metabolic phenotypes. Global ATGL-deficient mice accumulate TAG in most tissues, exhibit reduced plasma fatty acid concentrations, improved glucose tolerance, and insulin sensitivity, but die prematurely due to cardiomyopathy[4]. Rescue studies, expressing ATGL exclusively in the heart and thus preventing cardiomyopathy[5,6], protected mice from high-fat diet (HFD)-induced obesity[7,8]. Adipocyte-specific deletion of ATGL in mice[9] had similar molecular and phenotypic effects as observed in the rescue studies, implying that ATGL inhibition in adipose tissue leads to a

[1]Institute of Pharmaceutical Sciences, Pharmaceutical Chemistry, University of Graz, Graz, Austria. [2]Institute of Molecular Biosciences, Biochemistry, University of Graz, Graz, Austria. [3]Gottfried Schatz Research Center, Molecular Biology and Biochemistry, Medical University of Graz, Graz, Austria. [4]Field of Excellence BioHealth - University of Graz, Graz, Austria. [5]BioTechMed-Graz, Graz, Austria. ✉e-mail: ulrich.stelzl@uni-graz.at

beneficial metabolic phenotype. Similarly, pharmacological inhibition of ATGL with small molecule inhibitors represents a viable treatment strategy for obesity-associated disorders and cardiac insufficiency[8,10].

Activation of the lipolysis pathway in adipocytes occurs through β-adrenergic GPCR signaling for example by catecholamines. Lipolysis, including ATGL activity, is post-translationally regulated by proteins located at the LD surface[11]. The current model[2] proposes that under basal conditions members of the perilipin family coat the LD surface thereby restricting the access of ATGL to the LDs. At the same time, Perilipin 1 (PLIN1) sequesters the activating protein CGI-58 (gene name: *ABHD5*) preventing its association with and activation of ATGL. Upon stimulation, PKA-dependent phosphorylation of PLIN1 and CGI-58 leads to dissociation of the complex and CGI-58 release, which then binds and activates ATGL[12,13]. Additionally, PKA phosphorylation also recruits hormone-sensitive lipase (HSL), the second lipase required for DAG cleavage in the lipolysis pathway, from the cytosol to LDs[14]. However, this model is incomplete as studies showed that TG hydrolysis does not require PKA-dependent activation of HSL[15] and that a S239D phospho-mimicry variant of CGI-58 does not require PLIN1 phosphorylation for β-adrenergic stimulated ATGL activation[13]. Together, published data suggest that additional phosphorylation targets are elusive and that the order of molecular events in control of the lipolysis pathway is not fully understood. Detailed knowledge of protein-protein interaction (PPI) determinants is prerequisite for a full mechanistic understanding of ATGL regulation and ATGL-dependent lipolysis mechanism.

One of the most important ATGL PPI partners (Fig. 1A) is CGI-58, the co-factor required for full hydrolytic activity of ATGL[16]. Previously, structural protein domains required for the PPI have been identified[17–19], but specific ATGL amino acid residues or binding surfaces remained undetermined. Conversely, the interaction of G0S2 (G0/G1 switch gene 2) with ATGL potently inhibits ATGL TAG hydrolase activity[20]. Mutational analysis of a G0S2 peptide comprising amino acid residues 20 to 44 defined the minimal G0S2 binding site for the interaction with the N-terminal patatin domain of ATGL[20] (Fig. 1A). However, introducing single amino acid mutants in the full-length G0S2 protein did not disrupt the PPI in co-IP experiments and only slightly altered the inhibitory effect of G0S2 on ATGL activity, indicating that additional parts of G0S2 participate in this interaction[21]. Interaction determinants on the side of ATGL protein as well as the inhibitory mechanism are unknown. CIDEC (cell death-inducing DFFA-like effector c, also FSP27), exclusively found in white and brown adipose tissue, is another inhibitory protein of ATGL[22]. Interaction between ATGL and CIDEC inhibits ATGL-mediated lipolysis but does not directly affect ATGL's TAG catalytic activity[23,24]. The interaction is mediated via amino acid residues S120 to P220 of CIDEC[23] (Fig. 1A). Perilipins, such as Plin1 and Plin5, are major inhibitory regulators of lipolysis under basal conditions[25] by binding of CGI-58[12,26,27] and by restricting the access of lipases to the LD surface. Plin1 deficiency leads to increased lipolysis, hence Plin1 knock-out mice are lean, show reduced LD size, are resistant to HFD-induced obesity, and develop insulin resistance[28]. Interestingly, also upon overexpression of Plin1, adipose tissue is reduced and mice become resistant to HFD-induced obesity[29,30]. Like Plin1, Plin5 promotes TAG accumulation under basal condition but increases lipolysis in response to PKA stimulation[31,32]. Plin5 directly binds ATGL via residues R417 to F463 (Fig. 1A)[31,33]. Through its very C-terminus, Plin5 also directly interacts with mitochondria linking LDs to FA oxidation[31,34]. Heart-specific knockout of Plin5 in mice partially phenocopies ATGL loss of function[35,36].

Deep mutational scanning approaches that assess the functional effects of thousands of variants of a protein have become a prime tool in functional genomics[37]. They allow to study the impact of every possible single amino acid substitution in a protein on its function, protein stability, biophysical properties, and protein-protein interactions as well as a functional annotation of disease-associated human variants[38]. Yeast-based protein-protein interaction assays, such as yeast protein fragment complementation or yeast two-hybrid assays, that enable the enrichment of interacting protein variants or non-interacting variants, respectively, are key to deep mutational protein interaction scanning[39]. The assays have been used to scan several key protein interactions, including PPIs of the BBSome complex[40], the interaction between BARD and BRCA1[41], leucine zipper PPIs[42], the Ras-Raf pair[43,44], peptide binding of a SH3 and a PDZ domain[45] and a set of NF2 tumor suppressor protein interactions[46]. We apply reverse yeast two-hybrid (rY2H) growth selection for efficient enrichment of non-interacting mutant protein versions from comprehensive single amino acid protein variant libraries[40]. A quantitative sequencing readout identifies amino acid substitutions that negatively impact the capacity to interact with a protein partner (Fig. 1B). When assaying a deep mutational pool of a protein with multiple protein interaction partners, we can infer mutations that selectively impact the binding of individual partners rather than mutations that impair the function of the protein as a whole.

Here, we use the deep mutational interaction perturbation approach to comprehensively chart the impact of >4000 single amino acid substitutions in ATGL on the binding to its five most important interaction partners, CGI-58, G0S2, PLIN1, PLIN5, and CIDEC. The results of our deep mutational interaction scanning experiments first of all define contact sites for the interacting proteins that govern ATGL activity and function. Moreover, we specifically pinpoint a set of eleven switch mutations that selectively impair the interaction of ATGL with one of the five protein partners only. These switch mutations show defined functional patterns in vitro and lipolysis phenotypes in cells. Definition of amino acid resolution binding determinants for ATGL regulatory proteins provides mechanistic insights on the mechanism of regulated intracellular lipolysis.

## Results and discussion
### Deep mutational scanning interaction perturbation of ATGL-PPIs
We first established the interactions between full-length human ATGL and CGI-58, G0S2, Plin1, Plin5, and CIDEC in our Y2H assay system[47,48] (Fig. 1A). Applying the nicking mutagenesis protocol of Wrenbeck et al.[49], two libraries of ATGL variants were constructed. These two libraries contained mutations of each and every amino acid against alanine (A::GCT and A::GCA), lysine (K::AAA and K::AAG), glutamic acid (E::GAA and E::GAG), and leucine (L::TTG and L::CTA), using two different codons per amino acid exchange (Fig. 1B). This totaled up to 4000 single amino acid mutations in each ATGL cDNA library. During the subcloning into Y2H vectors, the colony number of the libraries was kept above 1 million, exceeding the theoretical number needed to maintain each ATGL variant in the library by 2 orders of magnitude. Sequencing of the two libraries confirmed that the number of mutations per codon and mutations per amino acid positions in ATGL did not change during the cloning procedure (library after mutagenesis vs after yeast mating, Fig. 1C). Read counts of non-programmed and programmed AKEL mutations between the two libraries were highly correlated with library 2 having a slightly higher rate of programmed mutations than library 1. Except amino acid position 1 no positional bias was observed (Fig. 1D).

The two ATGL variant libraries were each tested with the five wild-type interaction partner proteins in the perturbation screen (Fig. 1B). Additionally, wild-type protein ORFs were used in different Y2H vector configuration (Supplementary Data 1) so that two (Plin1, CIDEC), four (CGI-58, Plin5) and six (G0S2) screens were combined into one ATGL perturbation profile for each interaction partner (Fig. 2A). After sequencing, relatively low cut-offs were applied to generate the interaction profiles, which is reflected by many mutation hits distributed in the condensed profiles for the five proteins (Supplementary Data 2, Source data). Across the whole dataset, mutations of proline,

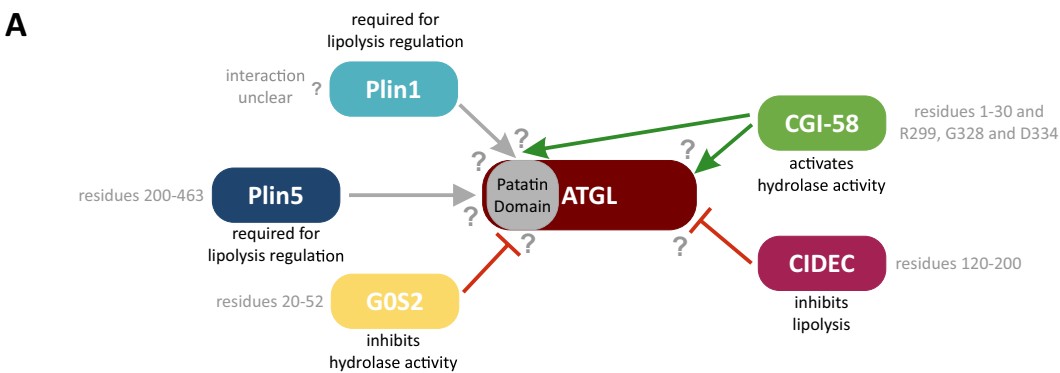

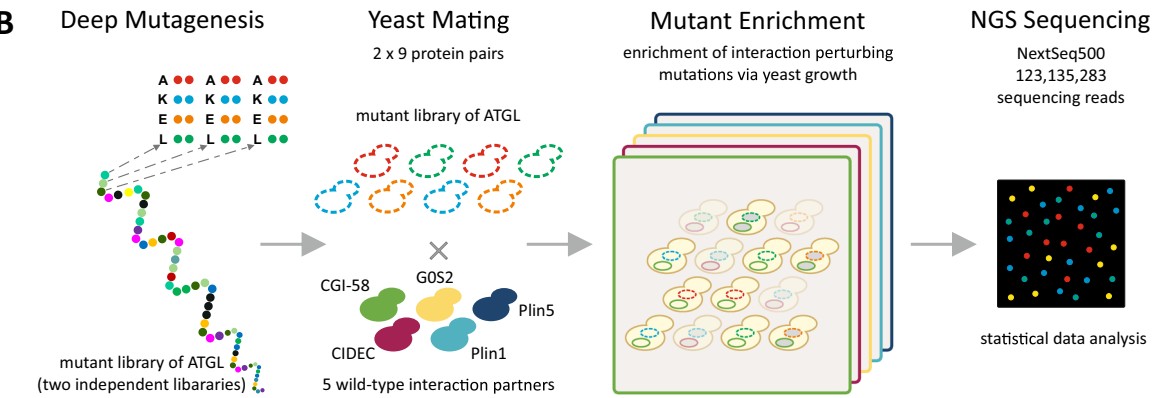

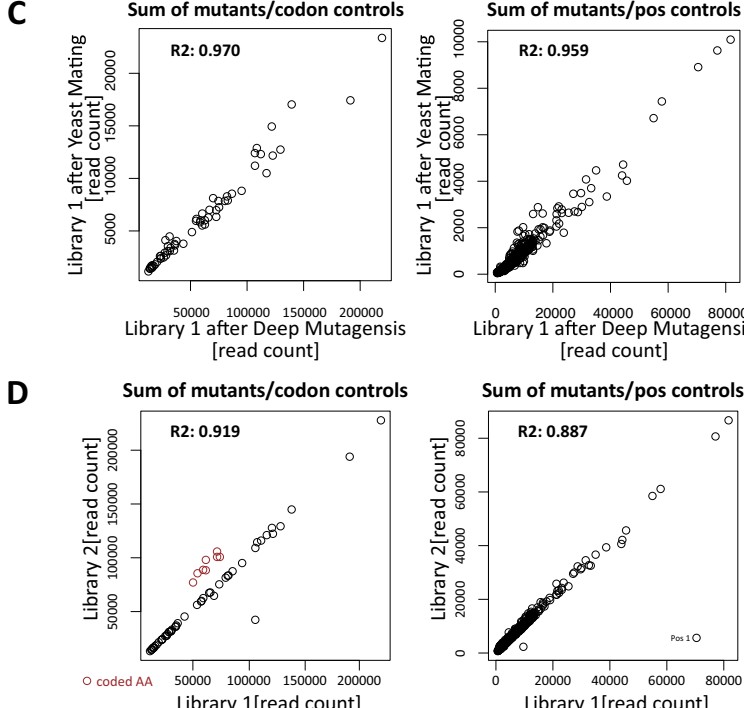

glycine, cysteine as well as threonine and serine contributed most frequently to the interaction-disrupting mutations, which may reflect the importance of those amino acids for folding and stability[42,45] (Fig. 2B). We detected many variants that impacted several protein interactions, however, mutations in the region around amino acid 250, a linker region between the patatin domain and the C-terminal part of ATGL, hardly had any impact on binding of the interaction partners. In

a close inspection of the profiles, we focused on identifying surface positions where mutations selectively impaired binding of one or two proteins, while having no or low enrichment values for the other interaction partners. For example, N39E had a high score for CGI-58 and G0S2 but not for the PLINs or CIDEC (Fig. 2C). P103K was enriched in the screen with Plin1 and Plin5, but not with the other three interaction partners. The mutational profile for L159 showed that, besides a

**Fig. 1 | Mapping ATGL protein interaction determinants using deep mutational interaction perturbation scanning. A** Schematic overview of ATGL and its interaction partners. Known AA residues mediating the interactions are annotated, see text for references. The Perilipin family members bind ATGL within the patatin domain. G0S2 also binds ATGL within the patatin domain, inhibits its hydrolytic activity. CIDEC, another lipolysis inhibiting protein interacts with the C-terminal part of ATGL. CGI-58 binds ATGL via the patatin domain as well as the C-terminal part. However, no interactions sites on ATGL for its partners were identified.
**B** Schematic overview of the deep mutational interaction perturbation screening approach. Plasmid libraries of ATGL were generated using array programmed deep mutagenesis, exchanging single amino acid to A, K, E, or L, respectively. Reverse Y2H strains were transformed with the ATGL libraries and mated with yeast strains

expressing a WT interaction partner. Interaction-disrupting mutations were enriched through yeast growth, and mutations perturbing the PPI were identified by NGS sequencing. **C** Scatter plot of read counts of library 1 in the Gateway Entry vector after deep mutagenesis with the library after yeast mating prior to growth selection. Read counts compared for sum of mutants per codon (left) as well as per position (right). R2 values indicate that the library did not change during the cloning procedure. **D** Scatter plot of read counts of the two independent ATGL libraries. Library 1 and Library 2 were compared for their sum of mutants observed per codon (left) as well as per position (right). Red colored circles indicate programmed AKEL mutations which are higher in library 2 than library 1. The outlier data point (Position 1) in library 1 in the right graph likely stems from an PCR amplification step early during preparation.

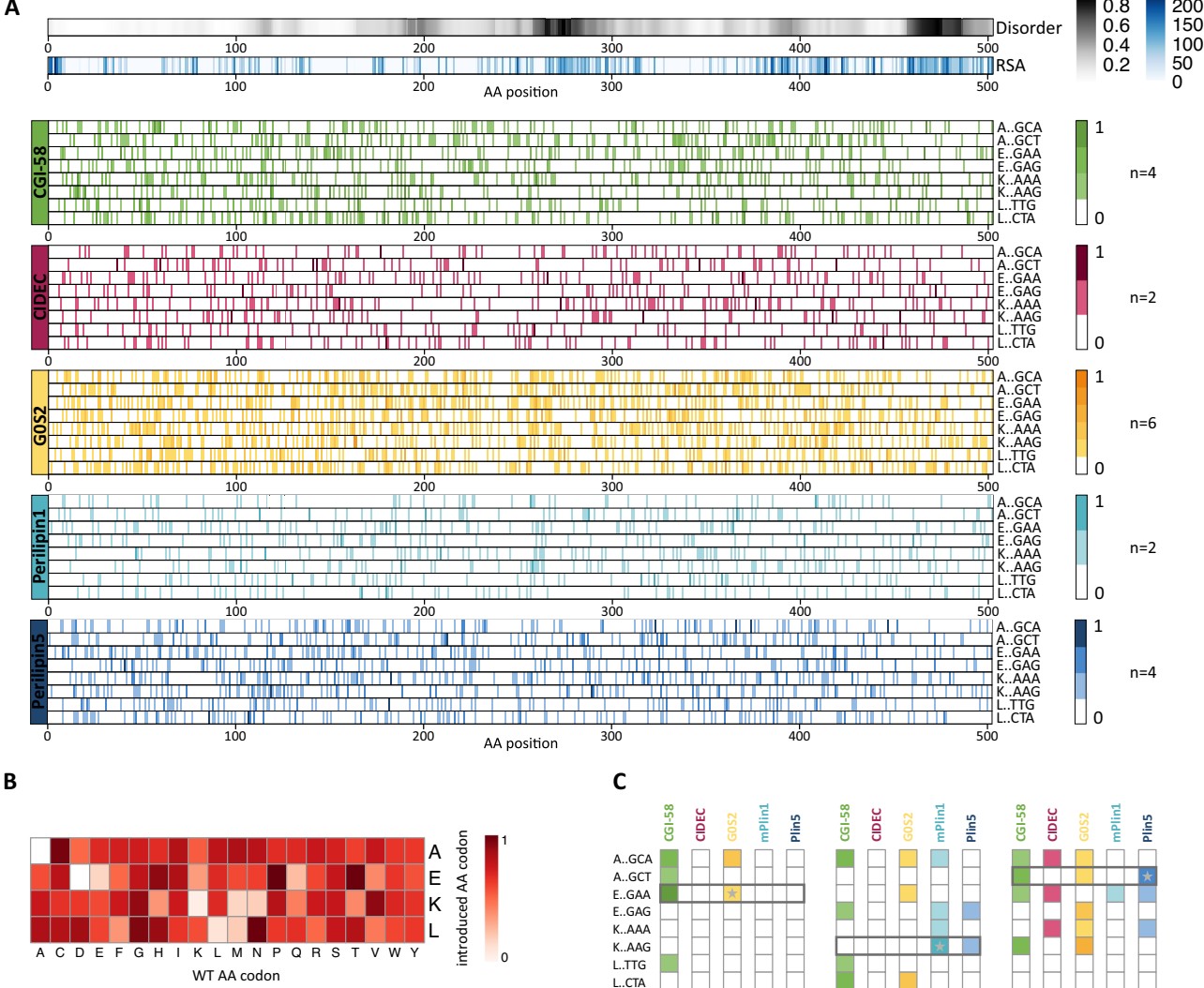

**Fig. 2 | Interaction perturbation maps of ATGL with five key regulatory binding partners. A** Combined interaction perturbation profiles for each ATGL binding partner. Top two profiles indicate relative solvent accessibility (RSA) and disorder obtained from IUPRED over the ATGL sequence. Combined, normalized individual profiles at codon resolution for the programmed AEKL mutations for the five interaction partners are shown (n number of experiments). Source data are provided as a Source Data file. **B** Heat map of frequency of single amino acid substitutions across all interactions. AKEL substitutions were counted for every amino acid across all probed interactions and normalized to the frequency of occurrence

in the ATGL wild-type sequence. **C** Zoom in at the perturbation profiles showing binding specificities of amino acid residues N39, P103, and L159. Mutations at position N39 affected the interaction of ATGL with CGI-58 and G0S2. The variant N39E (E::GAA) was selected as for further validation. Mutations at P103 influence several interactions with ATGL, however the exchange to K (K::AAG) selectively perturbed the interactions with Plin5 and Plin1. The L159A (A::GCT) substitution strongly perturbed the interaction with Plin5 and was selected for detailed variant analysis.

score for CGI-58 and G0S2, L159A most strongly reduced binding to Plin5. Evaluation of the variant profiles led to the prioritization of 52 ATGL variants for further individual assessment of binding behavior (Supplementary Data 3).

## Co-immunoprecipitation experiments define ATGL switch variants

Next, we tested the specificity of the mutational effects on individual protein interaction partners using a luminescence-based mammalian co-immunoprecipitation assay in HEK293T cells (LUMIER)[50,51]. Protein A was fused to the N-terminus of the ATGL variant protein set, comprising 52 single amino acid ATGL mutants as well as ATGL wild-type and a non-binding control protein. The Proteins were co-expressed with wild-type CGI-58, G0S2, CIDEC, Plin1, and Plin5, as firefly-luciferase-fusion proteins. The HEK293T cell lysates with the transiently expressed protein pairs were subject to immunoprecipitation in immunoglobulin-G coated microtiter plates and co-immunoprecipitation of the interaction partners was quantified as luminescence signal. Hence, we comprehensively tested all five interaction partners pairwise with 54 ATGL constructs in ~ 260 co-immunoprecipitation experiments for interaction in mammalian cells. To exclude that binding was substantially influenced by protein expression in the mammalian cell lines, similar expression levels of ATGL mutants was confirmed by western blot analysis (Supplementary Fig. 1). Normalized to wild-type ATGL binding, log2 fold change binding values were calculated from two independent experiments, performed in triplicates (Fig. 3A, Supplementary Data 4). Compared to wild-type ATGL, 29 variants did not show statistically significant differences in binding to any of the five interaction partners (Fig. 3B). This result is consistent with different assay sensitivities between the deep scanning sequencing readout and the co-immunoprecipitation experiments and reflects our attempts in prioritizing variants with highly selective effects on PPIs. However, the G189E mutant, while expressed as soluble protein of the correct size, perturbed the interaction with all interaction partners, an effect on binding considered non-informative. Importantly, 22 ATGL variants impacted individual protein interactions in a specific manner.

By virtue of its high binding signal, G0S2 was affected most strongly and by the highest number of ATGL variants, 12 in total. CIDEC on the other hand bound robustly to ATGL with only one mutation apart from G189E, namely P258L in the C-terminal part of ATGL, resulting in significantly decreased binding. This is in agreement with literature data that suggest the CIDEC binds to the C-terminal part of ATGL[23]. All other tested proteins had at least one interaction site in the patatin domain that also contains the catalytic lipase center (Figs. 1A and 3B). For CGI-58 we validated nine variants that decreased ATGL interactions. Interestingly, five of the mutations were located in the C-terminal part of the ATGL protein outside the patatin domain. We also validated 11 variants that impacted Plin1 and/or Plin5 interaction with ATGL. In particular, we considered ATGL mutations that disrupted the interaction with one partner only, ideally leaving the binding of all other unaffected. Such partner-specific ATGL mutations were termed interaction switch mutations, as they will perturb the protein interaction in a switch-like manner. They likely will have the most specific perturbation effect on ATGL regulation and signaling. We defined a set of 11 ATGL switch variants from our extensive binding data, three each for G0S2, Plin1, and Plin5 and two for CGI-58 (Fig. 3C).

The ATGL variants N39E, L62K, and L102K showed a strong decrease in binding of the inhibitory protein G0S2, while other interaction partners were hardly affected. The L102K variant decreased Plin1 and G0S2 binding, however the impact on G0S2 was much more substantial. Notably, Plin1-specific switch mutations P103K, A104E, and L178A, did not significantly impact Plin5 binding. The strongest reduction in binding of Plin5 was observed with the I70A, P86K, and L159A ATGL variants (Fig. 3C). The latter variant, L159A, also affected

binding of CGI-58 and showed the least specific perturbation pattern among the switch variants. The interaction with CGI-58 was disrupted by nine mutations in total, however, F348E and R351L were identified as two highly specific CGI-58 switch mutations in the C-terminal part of ATGL (Fig. 3C). Except for these two CGI-58 specific mutations, all switch mutations were located within the patatin domain of ATGL. In summary, we defined sets of two to three ATGL switch mutations for each of its canonical regulatory interaction partner, CGI-58, G0S2, Plin1, and Plin5.

## ATGL switch variant binding correlates with TAG hydrolase activity profiles

The identification of partner-specific ATGL interaction switch mutations implied that the protein variant is at least partially functional. Still, it cannot be excluded that the switch mutations have an impact on protein function, such as lipid hydrolase activity. To address this, TAG hydrolase (TGH) activity of the 11 single amino acid switch mutant proteins was assessed in vitro (Fig. 4).

Wild-type ATGL and mutant ATGL versions were expressed in Expi293 cells, a mammalian system optimized for protein expression. ATGL protein expression was monitored via western blot analysis showing comparably high expression levels of all variants in the soluble lysate (Fig. 4A). TGH activity of ATGL variants was determined in cell lysates with a PC/PI-emulsified radioactive isotope-labeled triolein substrate[52], and lysates of empty vector (EV)-transfected cells were used as negative control. Wild-type and all ATGL switch mutants were assayed for basal and CGI-58-stimulated hydrolase activity. Wild-type ATGL showed basal hydrolase activity that increased several-fold by stimulation with purified mouse CGI-58 protein (Fig. 4B) and cell lysates containing full-length human CGI-58 (Fig. 4C). Stimulation with either purified CGI-58 or cell lysates containing human CGI-58 resulted in the same activity patterns across all variants.

The TGH activities of mutant ATGL proteins can be grouped into, (1) inactive mutants, (2) mutants with reduced hydrolase activity, (3) mutants that had comparable activity as wild-type ATGL, also responding strongly to CGI-58 stimulation. Inactive mutants N39E, L62K, and L102K were insensitive to CGI-58 stimulation and the basal activities were indistinguishable from the empty vector control. The mutants I70A, P86K, and L159A had reduced hydrolase activity in comparison to wild-type ATGL and responded to CGI-58 stimulation by a moderate increase of TGH activity. The third group with TGH activity comparable with wild-type ATGL included P103K, A104E, L178A, F348E, and R351L. An intriguing observation was that the grouping of the ATGL mutants according to TGH activity correlated exactly with the type of switch mutants, i.e., the activity profiles traced to the impacted interaction partner. All three G0S2 switch mutants were inactive regardless of the presence of active CGI-58 (Fig. 4B, C, yellow label). The three Plin5 switch mutants, I70A, P86K, and L159A, showed no basal activity and a moderate increase when stimulated with CGI-58 (Fig. 4 B, C, dark blue label). The Plin1 switch mutants (Fig. 4B, C light blue label) and the CGI-58 switch mutant proteins (Fig. 4 B, C, green label) were active, however the latter showed moderately reduced stimulated activity compared to wild-type ATGL. Therefore, we tested the CGI-58 switch mutants, F348E and R351L, for dose-dependent stimulation by increasing the concentration of affinity chromatography enriched human CGI-58 (Fig. 4D). The experiment showed that basal TGH activity is not affected, however CGI-58 stimulation was impaired under saturated conditions. The TGH activity of a double mutant ATGL version (F348E + R351L) which combined the two CGI-58 switch mutations also responded to CGI-58 stimulation. It showed activity indistinguishable from the two single mutant versions in this in vitro assay (Fig. 4E). In summary, we grouped ATGL switch variants according to defined patterns of in vitro TGH activities. This activity grouping strikingly resembled the selective interaction losses of individual partner proteins of the ATGL switch mutations.

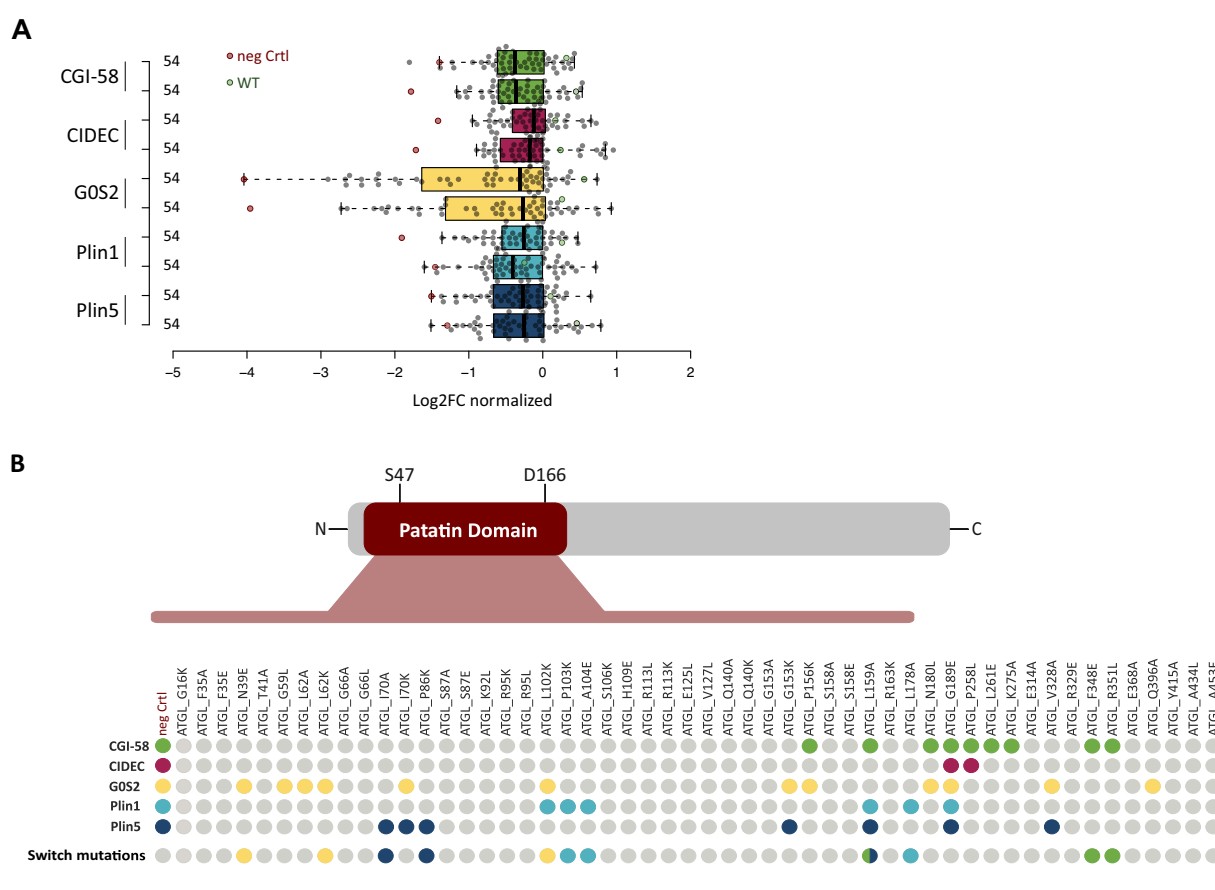

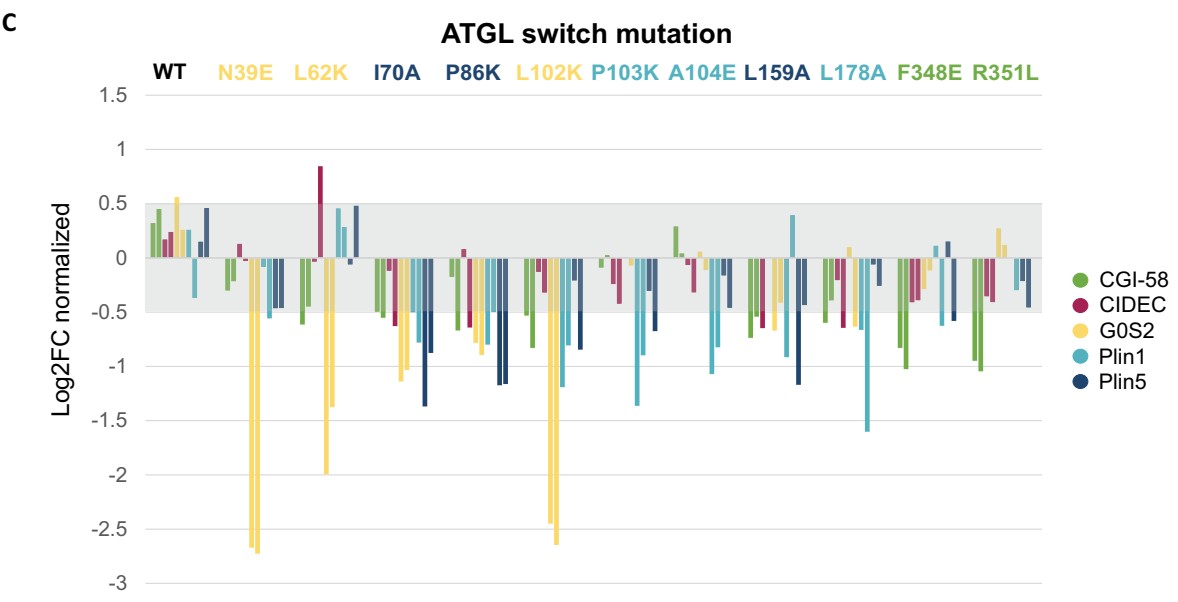

## Lipolysis activity of ATGL switch variant in cells

We next used live cell confocal fluorescence microscopy to assay the activity of ATGL switch variants in mammalian cells. HeLa cells were transfected with YFP-tagged ATGL variants and then loaded with oleic acid for 6–8 h, to induce LD formation. When stained with Bodipy, non-transfected cells showed a large number of LDs, while we readily observed distinct phenotypes of YFP-positive cells expressing switch

mutants (Fig. 5A). Cells that expressed wild-type ATGL did not stain for LDs suggesting that ATGL overexpression had high lipolytic activity in live cells associated with strongly reduced TAG storage. In contrast a very high fraction of cells (>94%) expressing either of the three G0S2 switch ATGL variants contained LDs similar to non-transfected cells. For each switch mutant, we quantified the number of YFP-positive cells for the presence of LDs reflecting lipolysis activity

**Fig. 3 | ATGL switch mutations selectively interrupt individual binding partners. A** Luciferase-based co-immunoprecipitation of 52 ATGL mutants with five protein interaction partners. Box plot of showing the mean of triplicate measurements of two experiments each probing murine Plin5 (blue), murine Plin1 (cyan), G0S2 (yellow), CIDEC (red), and CGI-58 (green) for interaction with the human ATGL variants (gray data points), ATGL wild-type (green data points) and a negative control (red data points). Log2 fold changes in binding relative to ATGL wild-type was determined within every experiment (Supplementary Data 4). Values were normalized across the experiments to the 3rd quartile indicating increased or decreased interaction signal. The boxes extend from 1st to 3rd quartile with the central band representing the median. The whiskers extend to the furthest point up to 1.5 times the interquartile range away from the nearest quartile. **B** Systematic overview of the binding data of the 52 ATGL variants. Variants are shown with their location within the patatin domain or in the C-terminal half of the protein. ATGL variants reducing the binding by more than twofold are marked as colored dots according to the affected interaction partner. The switch mutants are highlighted separately. Color code: CGI-58 green, CIDEC red, G0S2 yellow, mPlin1 cyan and mPlin5 blue. **C** Binding data for eleven ATGL switch mutations. For each interaction partner (color coded) normalized log2FC of two experiments performed in triplicates are shown. Single amino acid ATGL variants are color-coded according to the most specific binding alteration with a single partner, color code: CGI-58 green, CIDEC red, G0S2 yellow, mPlin1 cyan, and mPlin5 blue. Log2FC values within −0.5–0.5 (gray area) indicate wild-type-like binding behavior.

(Fig. 5B, Supplementary Data 5). The results obtained with the ATGL variants agreed well with our THG activity data in vitro. The G0S2 switch variants were inactive in vivo, and also the PLIN5 switch variants I70A and P86K showed a high number of LD-positive cells consistent with reduced in vitro activity. In contrast, ATGL PLIN5 switch variant L159A that showed reduced activity in the in vitro assay was active in vivo. The PLIN1 switch mutations showed full, wild-type-like activity in cells. The CGI-58 switch variants, F348E and R351L were largely active under these conditions, again consistent with the TGH results. However quantitatively, a fraction of 6–9% LD-containing cells indicated that the two switch variants with reduced binding of CGI-58 were functional, but exhibited reduced lipolytic activity. The hypothesis that a CGI-58 interaction site around amino acid position 350 in the C-terminal part of ATGL is critical for lipolysis is supported by assaying the ATGL variant that combined both CGI-58 switch variants in cells. Notably, the ATGL-F348E + R351L double mutant was inactive in the live cell lipolysis experiment (Fig. 5C) even though it was enzymatically active (Fig. 4E). Taken together, the live cell microscopy experiments revealed distinct cellular lipolysis phenotypes of ATGL variants which reflect the impact on the binding of ATGL to four interaction partners.

## Localization of switch variants to LDs

In order to test lipolysis activity in the presence of perilipins, we integrated human PLIN1 and PLIN5 into the genome of Flp-In T-REx™−293 cell lines. Induction of the expression of PLIN1 or PLIN5 led to large LDs upon oleic acid treatment (Supplementary Fig. 2). We then transfected the PLIN1 or PLIN5 expressing cell lines with wild-type ATGL and switch variants and observed that lipolysis was strongly inhibited even with fully active ATGL. This is in agreement with the literature[25,53], however, because of the strong lipolysis inhibition no differences in lipolysis activity were observed between any of the ATGL variants.

In cells expressing PLIN1 (Supplementary Figs. 3 and 5) or PLIN5 (Supplementary Figs. 4 and 6) we observed partial colocalization of ATGL and Bodipy-stained large LDs. In Hela cells or adipocytes, oleic acid treatment can result in cells with a very high LD content, with many large LDs (>1 μm) per cell. When these cells were fixed and stained with antibodies, ATGL accumulation was observed around the LDs in a rim-like appearance[54,55]. Here we used live cell confocal fluorescence microscopy of 293 cell lines stably expressing PLIN1 or PLIN5 which have an apparently lower LD content with fewer large LDs. Nevertheless, we observed partial colocalization on large LDs, in agreement with previous observations[12,17,54–56]. Compared to wild-type ATGL, cells transfected with one of the three G0S2 switch mutants showed less colocalization when considering the relative number of cells with clear colocalization signals. All PLIN5, PLIN1, and CGI-58 ATGL switch mutants showed partial colocalization with LDs, independent of the lipolysis activity observed in cells that do not express perilipin 1 or 5. In particular, the colocalization experiments showed that the two CGI-58 switch mutations and the double mutant ATGL-F348E + R351L localize to LDs similarly to wild-type ATGL

(Supplementary Figs. 3 and 4) and that the impairment of in vivo lipolysis activity of the CGI-58 double mutant ATGL variant is not due to differences in subcellular localization.

## ATGL variants inform about patient disease mutations

ATGL functional deficiency is associated with the rare neutral lipid storage disease with myopathy (NLSD-M)[57]. A few of the patients' ATGL variants were functionally characterized. For example, E172K, P195L or R221P have reduced lipase activity while G483R does not properly localize to LDs[56]. ClinVar[58] list 170 ATGL missense mutations found in patients with NLSD-M, all of which were classified as variants of unknown significance (VUS). Thirteen amino acid positions in the set of prioritized ATGL variants matched positions annotated in ClinVar. In contrast, we did not observe any overlap with population variants in ATGL[59] (Fig. 6A). Importantly, the ClinVar missense mutation overlap includes three of the switch mutants (P86K, A104E, and R351L), a result that sheds a light on their function (Fig. 6A). In our TGH assay, the P86K mutation was inactive under basal conditions, could only be stimulated partially and was inactive in cells, indicating that a substitution at this amino acid residue likely leads to an enzymatically impaired variant. While the amino acid substitutions A104E and R351L were enzymatically active, their interaction behavior is impaired, which likely results in altered ATGL function impacting NLSD. Taken together, these data demonstrate that single amino acid residues identified by deep mutational scanning can be useful to functionally annotate ATGL disease variants of unknown significance.

## ATGL variants inform lipolysis mechanism

ATGL switch mutants that prevent the binding of a specific binding partner were not necessarily close in primary sequence. For example, the G0S2 switch variants: N39E, L62K, and L102K span 60 amino acids in primary sequence. Likewise, Plin1 switch mutations, P103K, A104E, and L178A are more than 70 amino acids apart in primary sequence. Therefore, we projected the switch mutants on to available structural 3D models of ATGL (Fig. 6B). Three-dimensional structure predictions of ATGL position the catalytic residues S47 and D166 in the center of the N-terminal patatin domain[60] and additionally provide a low-confident model for the unstructured C-terminal part[61]. The three Plin1 switch positions are spatially clustered together pointing at a distinct interaction site, while the Plin5 switch positions I70 and P86 localize on a different side of the patatin fold (Fig. 6B). The Plin1 binding residues P103, A104 and L178 are surface residues with no intramolecular interactions of the side chains, in agreement with wild-type lipase activity of these ATGL variants. The side chains of Plin5 binding residues I70 and P86 were found to participate in intramolecular interactions (defined through a distance smaller than 5 Å to another amino acid), providing a rational for our results showing that the Plin5 switch variants are the least selective ones preventing the Plin5 interaction and reducing the catalytic activity of ATGL. L178A is a surface variant with substantially reduced catalytic activity in vitro. For reasons not fully resolved, the lipolysis activity of the variant was rescued under cellular conditions.

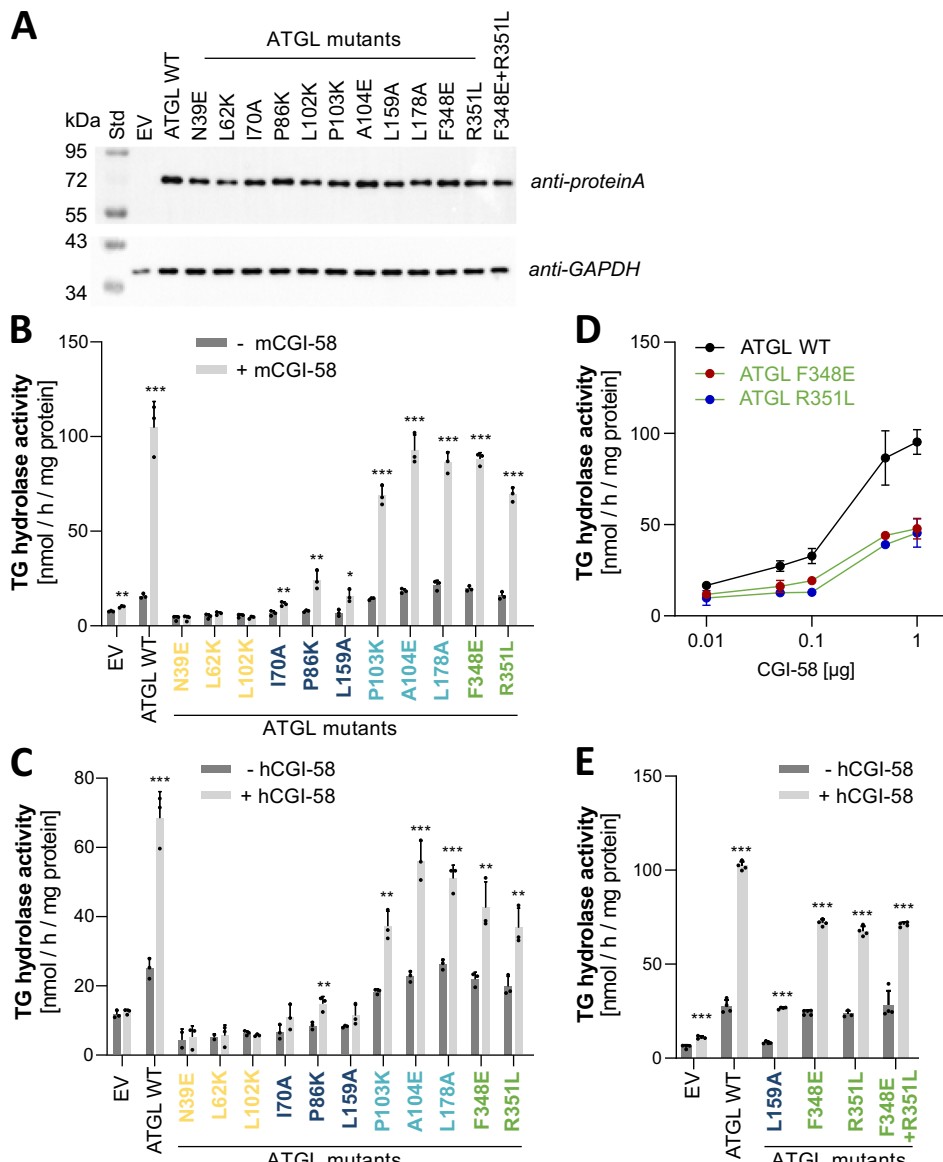

**Fig. 4 | Enzymatic activity of ATGL switch variants. A** Western blot analysis of ATGL switch variant expression in Expi293 cells. 5 µg protein of the whole cell lysate was separated on a 10% SDS-gel and stained with anti-Protein A Ab (top) and anti-GAPDH Ab (bottom). **B**, **C** TAG hydrolase activity detected in lysates of Expi293 cells. Activity was determined in the absence (−CGI-58, dark gray bars) or presence (+CGI-58, light gray bars) of purified full-length mouse CGI-58 from *E. coli* (**B**) or 10 µg lysate from cells expressing full-length human CGI-58 (**C**). EV, lysate with empty vector control, for measures of basal activity (−CGI-58, dark gray bars), the CGI-58 preparation was heat inactivated before addition. Single amino acid ATGL variants are color-coded according to the most specific binding alteration with a single partner, color code: CGI-58 green, G0S2 yellow, PLIN1 cyan, and PLIN5 blue. All samples were measured in triplicates and are shown as mean values + standard deviation. Statistical significance was determined by unpaired two-tailed *t*-test (*$p < 0.005$, **$p < 0.01$, ***$p < 0.001$). Source data are provided as a Source Data file. **D** Dose-dependent CGI-58 stimulation of ATGL switch variants. Data are presented as mean values, error bars indicate ± standard deviation of triplicate experiments. **E** TAG hydrolase activity detected in lysates of Expi293 cells as in D including the ATGL-F348E + R351L double mutant protein variant. ATGL-WT, L159A, F348E, and R351L are shown for direct comparison. Data are shown as mean values + standard deviation. Statistics as in (**B** and **C**).

Different ATGL binding determinants of the two perilipins are not unexpected. Plin5 is the major perilipin in the heart while Plin1 is highly expressed in adipose tissue. Functional studies of Plin1 and Plin5 in mice reveal different tissue-specific phenotypes[25,35,36,53,62]. Plin5 binds ATGL and CGI-58 via residues 200-463 in a mutually exclusive manner[33]. However, no ATGL residues important for perilipin binding are known. The binding mode of the ATGL-Plin1 interaction is elusive[25]. Using chimeric and mutant Plin5 and Plin1 protein constructs, the perilipins were shown to bind ATGL differentially. In contrast to Plin5, IF and FRET experiments in COS7 cells could not demonstrate colocalization or interaction of Plin1 and ATGL at LDs[33]. In our analysis, co-IP experiments in HEK293T cells established a stable interaction of

Plin1 and ATGL, that is characterized through three hydrolytically active Plin1 switch variants.

No ATGL binding determinants for the CGI-58 interaction are known either. We found variants affecting the interaction of ATGL with CGI-58 in the patatin domain as well as in the C-terminal part. Two CGI-58 switch mutations F348E and R351L were located in a structurally ill-defined C-terminal part of ATGL (Fig. 6B). In TGH activity measurements, the two switch variants showed wild-type basal activity, and somewhat reduced activity when stimulated with CGI-58 (Fig. 4D−E). In live cells, we observed a slightly increased fraction of LD-containing cells in comparison to wild-type ATGL (6−9%). Combining the two CGI-58 switch mutations resulted in an

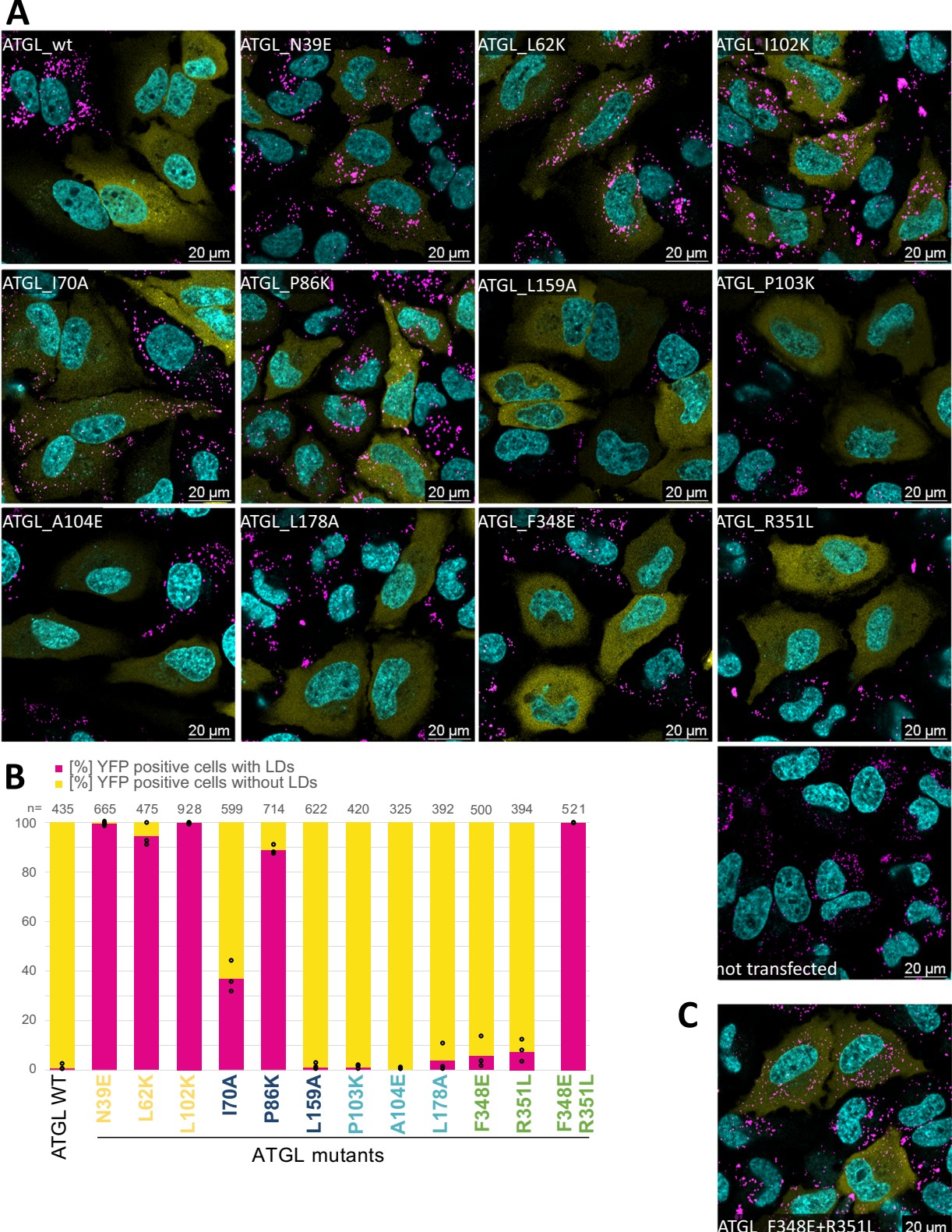

**Fig. 5 | ATGL variant activity in live cells. A** Confocal images of HeLa cells transfected with YFP-ATGL variants. Pictures are merged fluorescent images with Hoechst in cyan, YFP-ATGL in yellow, and Bodipy in magenta. Experiments were repeated independently three times with similar results. Transfected cell with LDs are counted in (**B**) with $n > 393$. **B** Fraction [%] of YFP-positive cells with LDs (magenta) and without LDs (yellow). Cells expressing YFP-ATGL from three independent experiments (black open dots) were classified according to presence of absence (<2) of LDs. $n$ = total number of cells counted. Single amino acid ATGL variants are color-coded according to the most specific binding alteration with a single partner, color code: CGI-58 green, G0S2 yellow, PLIN1 cyan, and PLIN5 blue. Source data are provided as a Source Data file. **C** Confocal image of HeLa cells transfected with YFP-ATGL-F348E + R351L double mutant. Picture as in (**A**). Experiments were repeated independently three times with similar results.

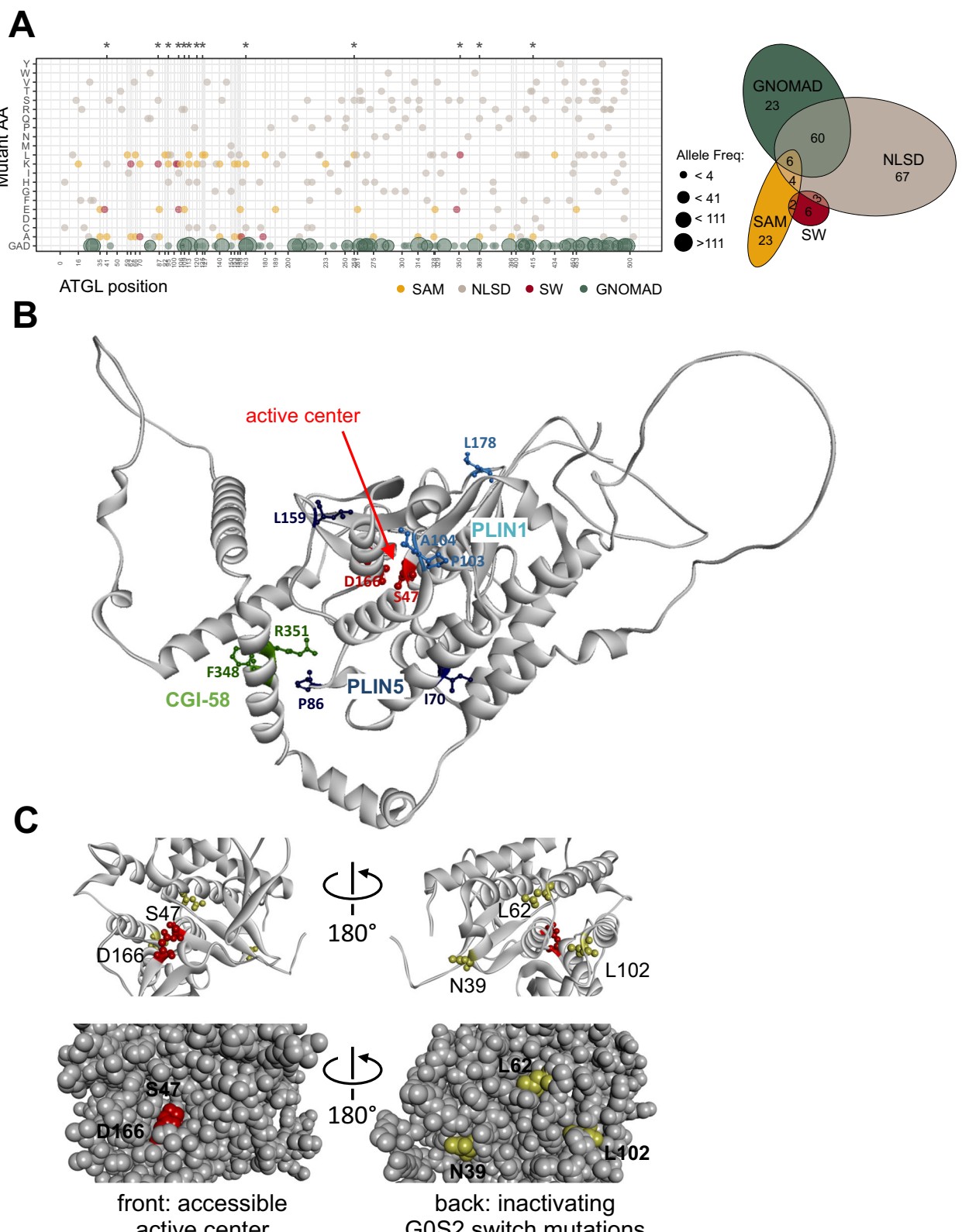

enzymatically active ATGL double mutant F348E + R351L variant, which however showed no lipolytic activity in cells. Localization of the double mutant CGI-58 ATGL variant to large LDs is not different to the single mutant or the wild-type protein. Therefore, we defined two highly specific CGI-58 switch variants outside of the patatin domain in the C-terminal part of ATGL involved in CGI-58 interaction and ATGL function. In combination, the two amino acid exchanges did not affect the enzymatic TGH activity but were strongly deleterious to ATGL lipolysis function.

It is well established that the N-terminal patatin domain extending to residue 254 of ATGL is sufficient for CGI-58-dependent stimulation of the lipid hydrolase activity[10,17,18,63]. The C-terminal part of ATGL harbors the LD targeting sequence[17,55] and phosphorylation sites including pT372, which abolishes LD localization[64] and pS406 which

**Fig. 6 | ATGL variants in a disease and structural context. A** Dotplot of annotated ATGL ClinVar missense substitutions associated with NLSD-M (ClinVar entries for PNPLA2, downloaded 10/2022). Gray dots: ClinVar mutations, yellow dots: selected ATGL mutations (SAM), red dots: switch mutations (SW). Green dots: variants from GnomAD (2.1.1. 03/2023). Asterisks on top highlight the positions T41, P86, R95, A104, H109, R113, R120, E125, R163, P258, R351, E368, and Y415, which overlap between the NLSD-M disease mutations and the SAM and SW mutants. *Right:* Euler diagram illustrating the overlaps of ClinVar missense mutations found in NLSD-M patients (gray), and the selected ATGL mutations (SAM, yellow) and the switch mutations (SW, red), and the missense variants from GnomAD (GNOMAD, green). **B** AlphaFold 3D structure model of the ATGL C-backbone [AF-E3VVS7] indicating binding partner contacts. Position and amino acid side chains (ball and stick) of (1) the Plin5 switch variants I70, P86, and L159 (blue) and (2) of the Plin1 switch variants P103, A104, and L178 (cyan) and (3) of the CGI-58 switch variants F348 and R351 are displayed. The catalytic dyad comprising S44 and D166 (red) are highlighting the active center within the patatin domain fold. **C** Zoom in of the patatin domain. *Left:* view through the hydrophobic cavity towards the catalytic dyad comprising S44 and D166 (red) and the G0S2 switch positions N39, L62, and L102 (yellow) in the back. *Right:* structure turned by 180° around a vertical axis providing a view from the back of the domain, the side where the G0S2 switch mutations are solvent exposed. Bottom left, space fill model showing open cavity to the catalytic center. Bottom right, 180° turned view as above with the three switch mutations exposed to the surface in the back of the domain. The active center is not accessible from the side of the G0S2 switch residues. G0S2 switch positions L62 and L102 are located closest (-15 Å) to the catalytic residue S47.

increases TG hydrolase activity[65]. Schweiger et al.[17] compared TGH activity of full-length ATGL protein with the truncated ATGL variant lacking 220 C-terminal amino acid residues. The truncated protein exhibited an up to 30-fold increased TGH activity upon CGI-58 mediated stimulation in vitro. Moreover, ATGL truncation increased binding to CGI-58 compared to full-length ATGL by an unknown mechanism in a GST pulldown ELISA assay, leading the conclusion that the C-terminal part is a critical regulatory site for CGI-58 function[17]. Gene mutations resulting in premature stop codons as well as missense mutations associated to ATGL malfunctioning cause NLSD-M (NLSD1). Notably, the missense mutations found in NLSD-M are distributed across the whole protein and not condensed in the N-terminus (Fig. 6A). Specifically, F348del and R351S point mutations, that affect the CGI-58 switch mutant positions characterized here, were found in patients with NLSD-M[58]. *ATGL* patient mutations include truncations of C-terminal parts of ATGL, which in principle leave the catalytically active patatin domain intact while impairing ATGL function: Exon 5 I212X, Exon 5 L255X, Exon 7 L318X, Exon 7 L319X, Exon 7 Q289X[66]. In vivo ATGL activity strongly depends on CGI-58 function as for example patient mutations in the CGI-58 gene (*ABHD5*) lead to a decrease in ATGL activity and result in TAG accumulation causing NLSD-I (NLSD 2)[66]. Also CGI-58 mutations, such as W21A/W25A, that change LD anchoring of the activator, impaired activation of the enzymatic activity of ATGL[67]. Overall, the functional and genetic data suggest that the 220 amino acid C-terminal part of ATGL has a crucial role for CGI-58 regulation of lipolysis activity. Here we defined switch residues in the region around amino acid position 350 that perturb the ATGL-CGI-58 interaction and are critical for cellular lipolysis. However, a full understanding of the role of the CGI-58 interaction with the C-terminus and the impact of the ATGL variants F348E and R351L in lipolysis requires further studies.

G0S2 is highly expressed in adipose tissue and differentiated adipocytes and constitutes the most potent inhibitor of ATGL activity[68,69]. G0S2 inhibits ATGL in a non-competitive manner[70] and functions independently of CGI-58[18,20], leaving the binding mode of G0S2 with ATGL undefined. We identified a series of ATGL variants that strongly reduce or abolish G0S2 binding, including three G0S2-specific switch variants: N39E, L62K, and L102K. Unexpectedly, all three were inactive in the TAG hydrolase activity assay, a finding that is strongly corroborated in our in vivo analysis. On the ATGL 3D structure model, these three residues were clustered together in proximity to the S44 and D166 catalytic dyad. The ATGL catalytic center is accessible through a cavity on the front side of the domain (Fig. 6C). All three G0S2 switch mutations however clustered on the surface of the back side of the domain from where the active center is non-accessible. (Fig. 6C, space fill model). This finding does not support a simple model where binding of G0S2 at or close to the substrate binding site directly perturbs substrate access or binding to the catalytic site. Rather it suggests that the GS02 switch mutations allosterically modulate the catalytic activity and that therefore G0S2 also inhibits the

enzymatic activity of ATGL allosterically through binding at the back side of the patatin domain.

As GS02 switch mutations and G0S2 binding to ATGL strongly inhibit hydrolase activity, we suggest that the amino acid exchanges and the inhibitor binding act allosterically, both impairing conformational flexibility of the enzyme. The findings are strongly reminiscent of the (auto)inhibition mechanism of protein-tyrosine kinases[71]. Protein kinases require conformational flexibility for phosphotransferase activity. Reduction of the conformational flexibility (both in its active or inactive conformation) through protein or small molecule binding inhibits kinase activity. For a large number of kinases this can be achieved through binding events at the back side of the kinase domain (e.g., SRC, ABL1), while the active site remains open and accessible[51,72]. Recently, a reversible and competitive small molecule inhibitor (NG-497) for human ATGL was described which bound in the cavity near the active site involving residues 60–146[10]. In agreement with our hypothesis that G0S2 binds the patatin domain on the back side, acting allosterically, the ATGL small molecule inhibitor and G0S2 were found to bind synergistically[10].

As the central protein with catalytic activity in lipolysis, ATGL interacts with a set of structurally diverse proteins. However, there is a lack of knowledge of ATGL binding determinants for its key regulatory interaction partners. To characterize the protein interactions with ATGL we applied comprehensive deep mutational interaction perturbation screening with five partner proteins. Mutations that affect binding to one protein typically have pleiotropic effects on binding to others[44,73]. However, here we defined a set of 11 switch mutations which are key in mediating interactions specifically to one of the partners with weak or negligible effects on the others. Our results reveal a number of insights concerning the regulation of ATGL function. First and foremost, we defined distinct binding determinants for G0S2, CGI-58, Plin1, and Plin5 at the amino acid level. The switch mutations can be useful experimental tools, providing most selective perturbations to test the functions of individual interaction partners in various biological contexts[74,75]. Second, we provide functional annotations for a set of 13 patient mutations associated to neutral lipid storage disease shedding light on their role in disease. Third, in agreement with their different functional roles in adipose and heart tissues, we identify distinct binding determinants for Plin1 and Plin5. Fourth, our results provide strong hints for the importance of the ATGL C-terminal part in the interaction with its activator CGI-58 and thus for cellular lipolysis. Fifth, the G0S2 switch variants of ATGL provide insight on the inhibition mechanism. The variants define an interaction site on the back of the patatin domain opposite of the catalytic center. The variants are no longer active suggesting an allosteric mode of action for G0S2 inhibition. It also implies that the development of drugs which allosterically modulate ATGL activity is feasible. In summary, this deep mutational scanning study provides amino acid resolution interaction surface information for ATGL's most important regulatory interaction partners.

## Methods

### Clones

ORFs were obtained in the Gateway Entry vector or were amplified from a cDNA clone via PCR and inserted to an entry vector via BP reaction. The LR reaction was used to clone ORFs in Y2H vectors (Prey: pACT4-DM and pCBDU-JW; Bait: pBTM116-DM and pBTMcC24-DM) and for LUMIER assay in protein A (pcDNA3.1PA-D57) and firefly-(pcDNA3.1V5-Fire) mammalian expression vectors[47,48]. Reactions were performed using a standard protocol from Invitrogen. Swissprot Sequence IDs: ATGL(Q96AD5); PLIN1 (O60240); mPlin1(Q8CGN5); PLIN5 (Q00G26); mPlin5(Q8BVZ1); G0S2(P27469), CIDEC(Q96AQ7).

### Mutagenic library construction

On-chip synthesized oligonucleotides were purchased from Cutsom Array, Inc. A total of 3692 ATGL primers were generated for each of the two libraries. The oligonucleotides (104 bp length: 60 bp primer + 44 bp adapter sequences) were amplified via PCR and adapter sequences were removed using restriction enzymes BciVI (New England BioLabs GmbH, R0596S) and BspQI (New England BioLabs GmbH, R0712S). Purified via an 8% acrylamide 8 M urea gel. Bands of 60 bp size were cut out and gel pieces were incubated in 200 μl dH2O at 4 °C for 24 h to resolve the 60 bp oligonucleotides.

Deep mutagenesis protocol from E. Wrenbeck et al. was used[49]. Minor changes were applied to increase the mutational power and reduce the WT background (Plasmid Safe digest and DPNI digest).

### Interaction perturbation reverse Y2H screen

To ensure a highly efficient yeast transformation and thus a high coverage of all mutants in the DNA libraries the lithium acetate/single-stranded carrier DNA/PEG method published by Gietz and Schießtl[76] was used. For one ATGL library four aliquots, with 1.5 μg plasmid DNA each, were prepared. After transformation procedure the aliquots of the same DNA libraries were united and 6 ml NB media were added. 2.5 ml of this suspension were plated on a large bioassay dish containing selective NB agar. The plates were incubated at 30 °C for 4 days. The total colony number was calculated by counting a 1 cm² area of grown colonies on 2 squares per plate.

$$Total\ colony\ number = \#colonies\ on\ 1cm^2 \times 580{,}81 cm^2 surface \times 4 plates$$

The yeast library transformants were used for the reverse Y2H protocol which was published by Woodsmith et al.[40]. and performed accordingly. After the growth selection of the mated yeast strains, the grown colonies were collected, lysed, and the DNA was purified through isopropanol precipitation and phenol-chloroform extraction and amplified via PCR.

### Data analysis

Next-generation sequencing (Illumina NextSeq 500(SN442), NextSeq High PE151) including barcoding of PCR samples was performed at the Sequencing Core Facility of the Max-Planck-Institute (Berlin, Germany). The obtained sequencing data were processed as described previously[40], a workflow incorporating Perl, as well as R scripts to analyze the paired-end sequencing data. In brief, fastq files of replicates were combined and converted to fasta files. The 150-mer reads were mapped to the wild-type ATGL cDNA sequence using the STAR Aligner. The amino acid codon enrichment was determined for each of the 18 protein interaction pairs tested.

$$Amino\ acid\ codon\ enrichment = \frac{observed\ total\ sequences\ for\ codon\ x}{expected\ total\ sequences\ for\ codon\ x}$$

The cutoff for the number of reads with mutations, as well as the fold-linear model enrichment cutoff were chosen in a library-specific manner (Supplementary Data 2). For non-coded mutations, the cutoff for the number of reads was 10× the median codon sequences across all positions. In a final step a custom perl script was used to detect and remove secondary mutations.

### Site-directed mutagenesis

The site-directed mutagenesis[77] was performed using the Phusion High Fidelity DNA Polymerase (New England BioLabs GmbH, E0553L). 3.5 ng template DNA as well as 5 μM of each primer were used. PCR was performed for 30 cycles with an annealing temperature of 65.5 °C. Subsequently, the PCR samples (10 μl) were digested with DPNI (0.2 μl New England BioLabs GmbH, R0176S) in CutSmart Buffer to remove the wild-type background. The reaction was inactivated at 80 °C for 20 min and competent TOP10 cells were transformed and utilized for plasmid preparation. Individual point mutations were verified via tag-sequencing.

### Luciferase-based co-immunoprecipitation experiments in mammalian cells

IgG-coated LUMITRAC 96 well plates: Wells were incubated with sheep gamma globulin ( Jackson ImmunoReasearch, 013-000-002) for 24 h at 4 °C, followed by blocking with BSA () for another 24 h at 4 °C. Finally, plates were incubated with AffiniPure Rabbit Anti-SheepIgG (H + L) ( Jackson ImmunoResearch, 313-005-003) for 24 h at 4 °C.

HEK293T (DMSZ ACC 635) cells were maintained in Dulbecco's Modified Eagle's Medium (DMEM) supplemented with 10% FBS and 1% Pen Strep at 37 °C, 7.5% CO₂ and 95% humidity. 2*10⁴ HEK293T cells in a well of a 96-well plate (coated with 0.05 mg/ml poly-D-lysin, Sigma Aldrich, P7405) were transiently transfected with 80 ng of a pcDNA3.1PA-D57 construct and 80 ng of a pFireV5-DM construct using PEI (1 mg/ml; Alfa Aesar, 900-98-6) as transfection reagent (ratio DNA:PEI = 1:5). After 48 h cell lysis was performed with SDS-free RIPA2 buffer (50 mM Tris HCl, 150 mM NaCl, 1 M EGTA, 1% NP-40, 0.25% sodium deoxycholate, 50 μg/ml dH₂O and 1 cOmplete™ mini-tablet per 10 ml buffer) for 45 min at 4 °C. Protein complexes precipitated from 80 μl cleared cell-extract in IgG-coated LUMITRAC plates for 2 h at 4 °C. Plates were washed tree times with 100 μl ice-cold PBS. 40 μl PBS and 40 μl Bright-Glo™ luciferase assay reagent (Promega, E2650) were added per well. The luminescent signal of firefly-luciferase activity was measured after ~10 min at RT in a micro-plate reader (Counter Beckmann DTX800). The assay was performed in triplicates, the mean log2 raw luminescence signals and standard deviation for each plasmid DNA pair was determined. Within every microtiter plate and for each tested luciferase-fusion construct log2 values were normalized to background pA-fusion protein or wild-type pA-fusion protein binding, respectively. Ratios larger than two over background or smaller than 0.5 when compared to wild-type ATGL and a z-score larger than two were considered. Protein expression was validated by western blotting.

### in vitro TG hydrolase assay

The ATGL variants were expressed in Expi293 (Thermo Fisher Scientific, A14635) cells. 2.5 × 10⁶ cells were transfected with 10.8 μg DNA using the ExpiFectamine™ 293 Transfection Kit (Thermo Fisher Scientific, A14525) according to manufacturer's instructions. Cells were harvested after 48 h, washed with PBS, resuspended in 4 ml HSL buffer (0.25 M sucrose, 1 mM EDTA, 1 mM DTT, 1 μg/ml pepstatin, 2 μg/ml antipain and 20 μg/ml leupeptin) and lysed by ultra-sonication (20% amplitude, 30 s). After centrifugation (20,000 × g, 4 °C for 10 min), the soluble extract was adjusted to a total protein level of 3.5 μg/μl. Protein expression (5 μg per lane) was validated by western blotting on PVDF using Abs against Protein A (1:3000, anti-rabbit-IgG-HRP-linked, NA934, GE Healthcare/Amersham) or GAPDH (1:10000, #2118 S, Cell signaling technology) and HRP-linked secondary antibody: anti-rabbit (1:10000, #7074, CST). The in vitro radiolabeled triglyceride hydrolase activity assay was performed according to Schweiger et al.[52].

## Live cell confocal fluorescence microscopy

HeLa S3 (ATCC CCL-2) cells were maintained in high-glucose +/+ DMEM (Dulbecco's Modified Eagle's Medium supplemented with 1% PEN-STREP [Penicillin-Streptomycin] and 10% FBS [fetal bovine serum]) and incubated at 37 °C, 7.5% CO2 and 95% humidity in a 24-well format (μ-Plate 24-Well Black ID 14 mm ibiTreat, ibidi, 221205/5). HeLa cells were transfected with YFP-ATGL variants using Polyethylenimine (PEI, linear, M. W. 20,000 [Alfa Aesar, 9002-98-6]) as transfection reagent (ratio DNA:PEI = 1:5). After 18 h incubation, cell were treated with 200 μM oleic acid (Na-Oleat:BSA = 3.25:1 in PBS) for 6–8 h. Cells were then stained with Hoechst 33342 (1:1000) and with Bodipy 493/503 (Thermo Fisher Scientific, 2256834; f.c. 0.2 μg/ml) or for 30 min.

T-REx™−293 (Invitrogen R71007) cells stably expressing doxycycline-inducible PLIN1 or PLIN5 (strep-HA-TO-FRT-N), respectively were constructed using the FLP-IN recombinational system according to standard procedures selecting the cells with 5 μg/ml Blasticidin and 50 μg/ml Hygromycin for 7 days. Cells were then maintained at 100 μg/ml Hygromycin[78]. For immunofluorescence microscopy of T-REx™−293 cells, stably expressing doxycycline-inducible PLIN1 or PLIN5 cells were transfected with YFP-ATGL variants using PolyJet™ (Signa Gen Laboratories, SL100688) as transfection reagent (ratio DNA:PolyJet = 1:3). Perilipin expression was induced with 2 μg/ml doxycycline (Fisher Scientific, #10224633). After 18 h incubation, cell were treated with 200 μM oleic acid (Na-Oleat:BSA = 3.25:1 in PBS) for 6–8 h. Cells were then stained with Hoechst 33342 (1:1000) and with Bodipy C12 558/568 (D3835, Fisher Scientific) f.c. 0.2 μg/ml for 30 min and analyzed via confocal microscopy.

HeLa cells were imaged on a Stellaris 5 confocal microscope (Leica) using an HC PL APO 63×, 1.4 NA objective. Excitation was for Hoechst at 405 nm, for Bodipy 493/503 at 500 nm or Bodipy C12 558/568 at 514 nm, and for YFP-ATGL at 516 nm. Emission was detected between 420 nm and 505 nm for Hoechst, 545 nm−625 nm for YFP-ATGL, and 505 nm−522 nm for Bodipy (493/503) or 560 nm−575 nm for Bodipy C12 (558/568) respectively. Images were analyzed with LASX software package (Leica, 3.7.6).

## Protein expression analysis

Sample preparation: 1.5 M cells were seeded and settle over-night and induced through addition of 1 μg/ml doxycycline (Fisher Scientific, #10224633) for 24 h. Protein extract was prepared in RIPA buffer (25 mM Tris pH 7.4, 150 mM NaCl, 1% NP-40 (v/v), 0.1% SDS (w/v), 0.5% Na-deoxycholate (w/v), 2 mM $MgCl_2$ + 1× Protease Inhibitor + 1× Phosphatase Inhibitor) utilizing additional sonication at 4 °C. 150 μg protein extract was alkylated, reduced, and digested with trypsin (1:50 wt/wt) and purified utilizing the S-Trap mini columns (Protifi), following the manufacturer's protocol. Peptides were lyophilized, resuspended in 0.1% formic acid, and diluted to a concentration of 200 ng/μl, whereby 1 μl was used per MS-injection.

Mass Spectrometry: Samples were analyzed on a timsTOF Pro ion mobility mass spectrometer (with PASEF® technology, Bruker Daltonics) in line with UltiMate 3000 RSLCnano UHPLC system (Thermo Scientific). Peptides were separated on a reversed-phase C18 Aurora column (25 cm × 75 μm) with an integrated CaptiveSpray Emitter (IonOpticks). Mobile phases A 0.1 vol% formic acid in water and B 0.1 vol% formic acid in ACN (Fisher Scientific, 10799704) with a flow rate of 300 nl/min, respectively. Fraction B was linearly increased from 2% to 25% in the 90 min gradient, increased to 40% for 10 min, and a further increase to 80% for 10 min, followed by re-equilibration. The spectra were recorded in DIA mode as previously and analyzed with DIA-NN v1.8.2. LFQ values were normalized to the total protein expression of each MS run (8 in total), log₂-transformed, and averaged across technical replicates (n = 2).

Protein expression was also validated by western blotting on PVDF using Abs against the HA tag (anti-HA.11 Epitope Tag antibody, 1:10000, Biolegend, 901501) and actin as loading control (anti-ß Actin (clone AC-15) antibody, 1:5000, Sigma, A5441).

## ClinVar data analysis

All annotated ClinVar entries for ATGL were downloaded (ClinVar entries for PNPLA2, downloaded 10/2022). The entries were filtered for single amino acid substitutions leading to a total of 172 annotated SNVs, all associated with NLSD-M.

## Graphics

All basic graphic illustrations, if not stated otherwise, were generated using R, RStudio, and the ggplot2 package.

## Statistics & reproducibility

Sequence analysis and statistical cut-offs are described in Woodsmith et al.[40]. Two-tailed unpaired *t*-test were performed as specified in the figure legends using GraphPad Prism. Z-score values were determined in luciferase-based co-immunoprecipitation experiments. No data were excluded from the analyses.

## Reporting summary

Further information on research design is available in the Nature Portfolio Reporting Summary linked to this article.

## Data availability

The AlphaFold model used to generate Fig. 6B, C is available under the accession AF-E3VVS7. ClinVar entries for PNPLA2, downloaded 10/2022 from https://www.ncbi.nlm.nih.gov/clinvar/?term=PNPLA2%5Bgene%5D&redir=gene. The sequencing data generated in this study have been deposited in the European Nucleotide Archive (ENA) under accession ENA Project Accession Number PRJEB60025. The mass spectrometry data generated in this study have been deposited via the ProteomeXchange PRIDE partner repository with the dataset identifier PXD049436. Source data are provided in the Source Data file. Source data are provided with this paper.

## Code availability

Code for sequence analysis were described in Woodsmith et al.[40].

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

## Acknowledgements

We thank Sandra Fasching for help with the experiments, Jonathan Woodsmith for support with data analyses, and Natalia Kunowska for critical reading of the manuscript. We thank Bernd Timmermann and the members of MPI-MG Sequencing Facility (Berlin) for performing the second-generation sequencing experiments. This research was funded in whole, or in part, by the Austrian Science Fund (FWF) [10.55776/DOC50] doc.fund Molecular Metabolism to U.S. The work was also supported by the Austrian Science Fund (FWF), project P30162 to U.S. and F73 SFB Lipid Hydrolysis to RZ, and the Field of Excellence BioHealth -University of Graz to U.S. and R.Z. The authors acknowledge the finan-cial support by the University of Graz.

## Author contributions

Conceptualization: U.S. Data curation: J.M.K., B.H. Formal Analysis: J.M.K., B.H. Funding acquisition: U.S., R.Z. Investigation: J.M.K., G.F.G., A.N., A.H., V.M., S.M., E.J-L. Methodology: J.M.K., G.F.G., H.E., R.Z., U.S. Project administration: Resources: U.S., R.Z. Software: Supervision: U.S., R.Z., H.E. Validation: J.M.K., G.F.G. Visualization: J.M.K., G.F.G., B.H., U.S. Writing – original draft: U.S., J.M.K. Writing – review & editing: U.S., J.M.K., and all authors.

## Competing interests

The authors declare no competing interests.
