## [Peer Review File · Nature Communications]

Mutational scanning pinpoints distinct binding sites of key ATGL regulators in lipolysisReviewer #1 (Remarks to the Author):

The manuscript by Johanna M. Kohlmayr et al. is a very thorough and important study of ATGL and its binding partners, providing novel insights into ATGL function and regulation. ATGL is the first lipase to mobilize triglycerides from lipid droplets, and is implicated in numerous human metabolic diseases as well as obesity and its associated pathologies. The major binding partners of ATGL are CGI-58, GOS2, PLIN1, PLIN5 and CIDEC. Here the authors carry out a state-of-the-art approach for mutagenesis of ATGL, and are able to identify mutants that specifically disrupt the interaction between ATGL and each of these 5 binding partners individually. They test the lipase activity of these mutants, both in vitro and in cells, and map their positions on 3D structures of ATGL. Binding determinants within ATGL were identified for Plin1 and CGI-58, the first time such interaction determinants have been demonstrated for these two partners of ATGL. Several other very interesting conclusions about ATGL function and regulation were revealed by this study, in particular that GOS2 inhibits the lipase activity of ATGL in an allosteric manner by binding to the back side of its patatin domain. In addition, the authors found that 13 patient mutations associated with neutral lipid storage disease with myopathy (NLSD-M) were identified in their mutagenesis screen, including 3 of their partner-specific "switch" mutations, shedding light on the mechanisms by which these mutations cause disease. I find this study to be of an exceptionally high quality both in terms of its conclusions and its experimental results. I have only a few minor comments for the authors to address.

Detailed comments:

1. Why do the authors use the term "switch mutation" for the mutations that abolish interaction with only one partner of ATGL? Usually "switch" means an exchange between two different states, whereas here, the mutations identified abolish one out of five partner interactions. The term "partner-specific mutation" would seem to me to describe the nature of the mutation more precisely, but perhaps I have missed something.
2. After identifying the "switch" mutations by co-immunoprecipitation assays, did the authors go back and test these mutations by classic yeast two-hybrid assays (co-transformation of two plasmids into the yeast two-hybrid strain)? This experiment is not required for the conclusions drawn, but as it is feasible, I just wanted to know if it was done.
3. Page 11, lines 1-3. The sentence "Plin5 is the major perilipin in the heart while Plin1 is highly expressed in adipose tissue, and knock-out as well as overexpression studies reveal different phenotypes" is not entirely clear. I assume the authors mean that knock-out and overexpression phenotypes of Plin1 are different than knock-out and overexpression phenotypes of Plin5, not that the knock-out and overexpression phenotypes are different (which is normally the case).

Reviewer #2 (Remarks to the Author):

ATGL is the rate-limiting enzyme catalyzing intracellular TG hydrolysis/lipolysis. It contains a patatin-like catalytic domain in the N-terminal region and a lipid droplet-targeting domain in the C-terminal portion. Post-translationally, ATGL is known to be regulated via interactions with multiple proteins including coactivator CGI-58, inhibitor GOS2 and lipid droplet coat proteins such as Plin1, Plin5 and Cidec. To elucidate the binding mode and sites in ATGL for protein interactions, the authors used a deep mutational scanning in combination with reverse yeast 2-hybrid growth selection to comprehensively profile single amino acid variants of ATGL that negatively affect the interaction with each of the aforementioned regulatory partners. Specific interaction perturbations subsequently were validated by coIP assays, and their impact on TG hydrolase activity and ability of ATGL to degrade cellular TG lipid droplets was also examined. Overall, this is an interesting and well executed and controlled study. The findings provide novel insight into the structural basis for the regulation of ATGL by individual partners, which may lead to a better understanding of how some of the ATGL variants in humans cause disease conditions such as NLSDM. The manuscript is, with only few exceptions, clearly written. However, there are several issues that when addressed, may help improve this manuscript from its present form.

Specific Points:

1. Given ATGL is regulated via complex mechanisms that involve multiple regulatory proteins, it is somewhat difficult to assess how some of the information obtained here would improve our understanding of ATGL regulation in vivo. In particular, when a specific amino acid variant of ATGL loses lipase activity and ability to degrade LDs, it may be limited in providing significant insight into how the partner the variant fails to interact with regulates ATGL in cells.
2. An existing model depicts that upon Plin phosphorylation and dissociation of Plin-CGI-58 complex, CGI-58 would be released at the LD surface to activate ATGL. For the ATGL variants deficient in binding to Plin1 (i.e. P103K, A104E and L178A) or Plin5 (i.e. P85K and L159A), would they behave differently in lipase assay or LD degradation assay when respective Plin proteins and/or CGI-58 are present?
3. Would ATGL variants deficient in Plin1 interaction respond differently to beta-adrenergic stimulation in adipocytes? Considering these variants can still be activated by CGI-58, information in this regard could shed new light on the involvement of Plin1 binding in the hormonal response of ATGL-mediated lipolysis.
4. Similarly, would ATGL variants deficient in Plin5 interaction respond differently to PKA activation in oxidative cell types?
5. Fig.5: in cells LD surface is a critical context in which ATGL exerts its lipolytic function. The question is, do any of the variants exhibit differences in LD localization?
6. The finding that F348E/R351L failed to degrade LDs is quite intriguing as either single mutation seems to increase the basal TG hydrolase activity. Is F348E/R351L catalytically active? What about its LD localization?
7. Given truncated ATGL lacking C-terminal 220 amino acids can still be activated by CGI-58, it is not immediately clear how interaction of CGI-58 with F348 and R351 of ATGL would contribute to coactivation.

Reviewer #3 (Remarks to the Author):

In this manuscript the authors examine the enzyme ATGL (Adipose Triglyceride Lipase, central in intracellular lipolysis), identifying key sites involved in its interaction with important regulatory partners. Their approach, involving deep mutational scanning, co-immunoprecipitation and enzymatic assays provides new insights into the regulation of lipolysis and disease-associated mechanisms in cells.

Minor Comments

1. The full name of ATGL should be provided once in the Abstract.
2. Figure 1C,D axis labels should specify the units (i.e., Read Count)
3. Why does Supplementary Table 1 include mention of Plin2 and Plin3? Were these also screened? Also, what does the 'm' designation in front of the names in some of the WT column entries represent? Is this mouse? If so, why were both mouse and human used?
4. In general, please provide a better legend and explanation for everything presented in the Supplementary Tables. While most things can be figured out, it would make for an easier read if everything was clearly defined up front.
5. The authors should provide a better explanation of how they prioritized the ATGL variants selected for further study (listed in Supplementary Table 4).
6. On Page 7, Lines 18/19 the authors mention that CIDEC can serve a positive control. I don't agree with this statement (which the authors don't really justify in detail) and think it is best removed.
7. The authors should provide a more standard loading control (e.g., Tubulin) for the Western blot presented in Figure 4A rather than Ponceau S staining.

8. The authors should include an image of non-transfected cells in Figure 5A.
9. English headers should be used in Supplementary Table 6.
10. The authors should better discuss the observation that the L159A mutant showed reduced activity in vitro but was highly active in vivo.

Major Comments

1. There seems to be an issue with consistency of effect for codon substitutions resulting in the same amino acid change. For example, in Figure 2C the two L159A codon substitutions presented have different profiles (i.e., one affects CIDEK, the other PLIN5)? This appears to be relatively widespread. Do the authors have an explanation for this?
2. The authors mention that before performing LUMIER they verified that all ATGL mutants were similarly expressed using Western Blot, however I don't see these blots presented anywhere. Please include this data.
3. I'm unclear on the data presented in Figure 3C. The authors mention they are looking at 'switch' mutations that only affect interaction with a single partner, however the data presented in Figure 3C suggests that some of these mutants still affect interaction reasonably well with more than one partner (e.g., I70A, P86K, L102K). The authors do touch on this briefly in the last paragraph of page 7, but I feel they are understating the level of effect. I think the authors should be more up front about describing these mutants as affecting certain interactions more strongly, as opposed to saying other interactions are 'unaffected'.

REVIEWER COMMENTS

Reviewer #1 (Remarks to the Author):

The manuscript by Johanna M. Kohlmayr et al. is a very thorough and important study of ATGL and its binding partners, providing novel insights into ATGL function and regulation. ATGL is the first lipase to mobilize triglycerides from lipid droplets, and is implicated in numerous human metabolic diseases as well as obesity and its associated pathologies. The major binding partners of ATGL are CGI-58, GOS2, PLIN1, PLIN5 and CIDEC. Here the authors carry out a state-of-the-art approach for mutagenesis of ATGL, and are able to identify mutants that specifically disrupt the interaction between ATGL and each of these 5 binding partners individually. They test the lipase activity of these mutants, both in vitro and in cells, and map their positions on 3D structures of ATGL. Binding determinants within ATGL were identified for Plin1 and CGI-58, the first time such interaction determinants have been demonstrated for these two partners of ATGL. Several other very interesting conclusions about ATGL function and regulation were revealed by this study, in particular that GOS2 inhibits the lipase activity of ATGL in an allosteric manner by binding to the back side of its patatin domain. In addition, the authors found that 13 patient mutations associated with neutral lipid storage disease with myopathy (NLSM) were identified in their mutagenesis screen, including 3 of their partner-specific “switch” mutations, shedding light on the mechanisms by which these mutations cause disease. I find this study to be of an exceptionally high quality both in terms of its conclusions and its experimental results. I have only a few minor comments for the authors to address.

We thank the reviewer for her/his positive assessment of our work.

Detailed comments:

1. Why do the authors use the term “switch mutation” for the mutations that abolish interaction with only one partner of ATGL? Usually “switch” means an exchange between two different states, whereas here, the mutations identified abolish one out of five partner interactions. The term “partner-specific mutation” would seem to me to describe the nature of the mutation more precisely, but perhaps I have missed something.

Many mutations have rather pleiotropic effects on multiple binding partners e.g. through perturbation of structure or protein levels etc.. We want to elucidate partner-specific mutations and termed them switch mutations with reference to turning an interaction on and off (that would indeed be two different stages). In our interaction centric view, we mean on/off switch with respect to an interaction. When introducing the switch mutation term, we now also refer to them as partner-specific mutations and refer to the interaction switch (page 7, second paragraph).

2. After identifying the “switch” mutations by co-immunoprecipitation assays, did the authors go back and test these mutations by classic yeast two-hybrid assays (co-transformation of two plasmids into the yeast two-hybrid strain)? This experiment is not required for the conclusions drawn, but as it is feasible, I just wanted to know if it was done.

No, we did not do this specific Y2H experiment. Co-immunoprecipitation in HEK293 cells is an orthogonal approach for validation of the effects of mutations on the interaction between the mammalian proteins and therefore likely more biologically relevant.

3. Page 11, lines 1-3. The sentence “Plin5 is the major perilipin in the heart while Plin1 is highly expressed in adipose tissue, and knock-out as well as overexpression studies reveal different phenotypes” is not entirely clear. I assume the authors mean that knock-out and overexpression phenotypes of Plin1 are

different than knock-out and overexpression phenotypes of Plin5, not that the knock-out and overexpression phenotypes are different (which is normally the case).

We have amended the sentence to be clearer in which phenotypes were reported in the literature (page 11, second paragraph).

Reviewer #2 (Remarks to the Author):

ATGL is the rate-limiting enzyme catalyzing intracellular TG hydrolysis/lipolysis. It contains a patatin-like catalytic domain in the N-terminal region and a lipid droplet-targeting domain in the C-terminal portion. Post-translationally, ATGL is known to be regulated via interactions with multiple proteins including coactivator CGI-58, inhibitor GOS2 and lipid droplet coat proteins such as Plin1, Plin5 and Cidec. To elucidate the binding mode and sites in ATGL for protein interactions, the authors used a deep mutational scanning in combination with reverse yeast 2-hybrid growth selection to comprehensively profile single amino acid variants of ATGL that negatively affect the interaction with each of the aforementioned regulatory partners. Specific interaction perturbations subsequently were validated by coIP assays, and their impact on TG hydrolase activity and ability of ATGL to degrade cellular TG lipid droplets was also examined. Overall, this is an interesting and well executed and controlled study. The findings provide novel insight into the structural basis for the regulation of ATGL by individual partners, which may lead to a better understanding of how some of the ATGL variants in humans cause disease conditions such as NLSDM. The manuscript is, with only few exceptions, clearly written. However, there are several issues that when addressed, may help improve this manuscript from its present form.

We thank the reviewer for the interest in our work.

Specific Points:

1. Given ATGL is regulated via complex mechanisms that involve multiple regulatory proteins, it is somewhat difficult to assess how some of the information obtained here would improve our understanding of ATGL regulation *in vivo*. In particular, when a specific amino acid variant of ATGL loses lipase activity and ability to degrade LDs, it may be limited in providing significant insight into how the partner the variant fails to interact with regulates ATGL in cells.

We agree with the reviewer that for functional studies inactive variants are less useful than others. However, we clearly define groups of variants where interaction perturbation corresponds to the enzymatic activity *in vitro*. We also show that *in vitro* activity may not correspond to *in vivo* activity of variants with the most notable example of the CGI-58 switch double variant that had impaired lipolysis (new data for the revision).

We believe that this comment essentially refers to the GOS2 switch variants and agree that these variants are less informative for *in vivo* studies than others. Nevertheless, the result as such is unexpected, as mutations that prevent the interaction with ATGL's high affinity inhibitor are inactive! As described in the manuscript it allows a strong, novel hypothesis towards the mechanism of GOS2 inhibition.

2. An existing model depicts that upon Plin phosphorylation and dissociation of Plin-CGI-58 complex, CGI-58 would be released at the LD surface to activate ATGL. For the ATGL variants deficient in binding to Plin1 (i.e. P103K, A104E and L178A) or Plin5 (i.e. P85K and L159A), would they behave differently in lipase assay or LD degradation assay when respective Plin proteins and/or CGI-58 are present?

The question if Plin1/5-switch ATGL variants would behave differently in the presence of perilipins is an excellent one. It also strongly relates to point 3 and point 4, as it is well established that Plin1 plays a major role in adipocytes and Plin5 in heart decorating intracellular lipid droplets in oxidative tissues. Plin1 deficiency results in decreased lipid storage capacity in fat cells and Plin5 deficiency is associated with increased reactive oxygen species formation, impaired mitochondrial function, and cardiac myopathy.

Therefore, we want to address points 2-4 together. We would like to argue that assaying the activity of ATGL variants and the respective phenotypes in differentiated adipocytes or cultured cardiomyocytes is not within the scope of this revisions. However, to address the influence of perilipins on lipolysis, we created stable, inducible PLIN1 and PLIN5 T-REx™-293 cell lines. We monitored the expression of PLIN1 and PLIN5 (new Suppl. Figure 2). We then transiently transfected ATGL switch variants and, after oleic acid treatment, assessed the lipolysis activity through confocal microscopy (as in Figure 5). An analogous experiment was performed using PLIN-ATGL co-transfection.

In agreement with the literature, expression of PLIN1 and PLIN5 strongly reduces lipolysis so that we do not observe substantial activity for wild type or other active ATGL variants in comparison to the inactive variants (Figure below, total number of YFP positive T-REx™-293 cells per sample was between 551-1355). Therefore, different, advanced experimental setups will have to be employed to investigate ATGL switch variants in different specific biological context and establishing such systems is beyond the scope of this work.

3. Would ATGL variants deficient in Plin1 interaction respond differently to beta-adrenergic stimulation in adipocytes? Considering these variants can still be activated by CGI-58, information in this regard could shed new light on the involvement of Plin1 binding in the hormonal response of ATGL-mediated lipolysis.

See answer to 2.

4. Similarly, would ATGL variants deficient in Plin5 interaction respond differently to PKA activation in oxidative cell types?

See answer to 2.

5. Fig.5: in cells LD surface is a critical context in which ATGL exerts its lipolytic function. The question is, do any of the variants exhibit differences in LD localization?

This question refers to the experiments shown in Figure 5, where transfected cells show evenly distribution of ATGL in the cytoplasm. Active ATGL variants hardly have droplets while inactive do. We always assess the LD content in transfected cells while the non-transfected cells with LDs serve as internal control. In this scenario it is prohibitively difficult to assess co-localization, as for the active versions the droplets are gone (therefore localization should not be impaired) and for the inactive ATGL versions, the fluorescence is distributed and droplets are small puncta. To assay LD localization, and observe ATGL accumulation at LDs we need to inhibit lipolysis and create more stable lipid droplets.

In our experiments addressing point 2 through expression of PLIN1 or PLIN5, we inhibited lipolysis which led to larger droplets. Indeed, in these experiments we observe colocalization of YFP-tagged ATGL variants on large perilipin coated LDs. In agreement with the literature (Schweiger 2008, [10.1074/jbc.M710566200]; Tavian 2012 [10.1093/hmg/dd3388]) only partial co-localization of ATGL and Bodipy stained LDs was observed. The data are presented in a new Suppl Figure 2 and described on page 9 to 10.

G0S2 switch variants (enzymatically inactive) show reduced colocalization in both PLIN1 and PLIN5 overexpressing cells. All PLIN5, PLIN1 and CGI-58 switch mutants showed partial co-localization with large LDs, independent of the lipolysis activity observed in cells that do not express perilipin 1 or 5.

CGI-58 switch variants, including both the single mutants F348E and R351L that are active as well as the F348E+R351L double mutant that is inactive, show colocalization in PLIN1 and in PLIN5 overexpressing cells, comparable to wild type like colocalization. The co-localization experiments clearly showed that the impairment of *in vivo* lipolysis activity of the CGI-58 double mutant variant is not due to differences in subcellular localization. (see next point)

6. The finding that F348E/R351L failed to degrade LDs is quite intriguing as either single mutation seems to increase the basal TG hydrolase activity. Is F348E/R351L catalytically active? What about its LD localization?

We agree that our finding that the F348E/R351L double mutant is inactive in cells is intriguing. We addressed the reviewer's question by performing *in vitro* activity assays and obtained a clear result. The F348E/R351L double mutant is active *in vitro*, indistinguishable from the individual single mutant variants. In light of the literature which shows that the C-terminal part in principle is dispensable for enzymatic activity on an artificial substrate *in vitro*, and our results with the single mutant ATGL versions, this is expected. This experiment strengthens our previous finding demonstrating that the C-terminal region around amino acid position 350 is an ATGL-CGI-58 interaction site and critical for cellular lipolysis (discussed on page 11-122). The C-terminal part of ATGL, specifically the interaction region around aa 350, is essential for lipolysis *in vivo*. We also note that in experiments with the perilipin expressing cell lines (see point 5), where lipolysis is strongly reduced, we observe co-localization of the F348E+R351L ATGL variant on large LDs, similar to the extent of colocalization with the wild type ATGL.

7. Given truncated ATGL lacking C-terminal 220 amino acids can still be activated by CGI-58, it is not immediately clear how interaction of CGI-58 with F348 and R351 of ATGL would contribute to coactivation.

We agree with the reviewer that there are open mechanistic questions that relate to the CGI-58 dependent activation of ATGL. We tried to address these issues in the discussion and also provide literature support showing the importance of this region for ATGL activation. We show that there are biochemically data in the literature that support the importance of the C-terminal part for regulation of ATGL activity, and add details on genetic data from NSLD-M patients strongly supporting the functional importance of this region. NSLD-M is thought to be caused by impaired ATGL activity. Several truncation mutations that keep the patatin domain intact were found in patients. Importantly, F348del and R351S point mutations, that affect the CGI-58 switch mutant residues characterized here, were found in patients with NSLD-M. However, we also state clearly that further studies in different specific biological context are required for a full mechanistic understanding of lipolysis regulation (see discussion, page 12).

Reviewer #3 (Remarks to the Author):

In this manuscript the authors examine the enzyme ATGL (Adipose Triglyceride Lipase, central in intracellular lipolysis), identifying key sites involved in its interaction with important regulatory partners. Their approach, involving deep mutational scanning, co-immunoprecipitation and enzymatic assays provides new insights into the regulation of lipolysis and disease-associated mechanisms in cells.

Minor Comments

1. The full name of ATGL should be provided once in the Abstract.

Ok done.

2. Figure 1C, D axis labels should specify the units (i.e., Read Count)

Ok done.

3. Why does Supplementary Table 1 include mention of Plin2 and Plin3? Were these also screened? Also, what does the 'm' designation in front of the names in some of the WT column entries represent? Is this mouse? If so, why were both mouse and human used?

We thank the reviewer for checking Table 1 carefully.

Indeed, some lines contained wrong entries that should have been omitted, as they were not included in the analysis at all. We corrected the table. We note that the data upload is correct (PRJEB60025).

We added legends for the Suppl. Tables describing each column (see next point), and also explain in the legend that in some of the screening experiments mouse ORFs were used. However, they are highly conserved and no position with a mouse / human variation was considered.

4. In general, please provide a better legend and explanation for everything presented in the Supplementary Tables. While most things can be figured out, it would make for an easier read if everything was clearly defined up front.

Ok, done: A column by column legend is provided on the legend sheet.

5. The authors should provide a better explanation of how they prioritized the ATGL variants selected for further study (listed in Supplementary Table 4).

As explained on page 6, first paragraph, we did not prioritize mutations aiming at the strongest effects, rather we looked for robust signals, e.g. found more than once, that nevertheless showed up at positions where mutations selectively impaired binding of one or two proteins, while having little effect on the other interaction partners. Many positions with enrichment signal from our screen will have pleiotropic effects on several interaction partners when mutated.

6. On Page 7, Lines 18/19 the authors mention that CIDEC can serve a positive control. I don't agree with this statement (which the authors don't really justify in detail) and think it is best removed.

We agree with this point and deleted this sentence.

7. The authors should provide a more standard loading control (e.g., Tubulin) for the Western blot presented in Figure 4A rather than Ponceau S staining.

Because we did not keep the membranes as such, we re-blotted the extracts and provide new blots and a loading control (GAPDH). The blot for the standard loading control is now included in Figure 4A.

8. The authors should include an image of non-transfected cells in Figure 5A.

We included the experiment with non-transfected cells in Figure 5. We agree with reviewer that non-transfected cell show the extend of LD formation after oleic acid treatment. Please note that in every image section we also included non-transfected cell for internal control/comparison purposes.

9. English headers should be used in Supplementary Table 6.

Yes done, we apologize for this oversight.

10. The authors should better discuss the observation that the L159A mutant showed reduced activity *in vitro* but was highly active *in vivo*.

We understand that this variant shows somewhat inconsistent results across the assays, and note that we do not have a final mechanistic answer to this. It is clear from the co-IP pattern that this variant has the "least specific" perturbation effect among the switch variants (amended text page 7, third paragraph) which could be a reason for the differences between *in vitro* and *in vivo* results. We improved our discussion on page 11, end of first paragraph pointing at this observation.

Major Comments

1. There seems to be an issue with consistency of effect for codon substitutions resulting in the same amino acid change. For example, in Figure 2C the two L159A codon substitutions presented have different profiles (i.e., one affects CIDEC, the other PLIN5)? This appears to be relatively widespread. Do the authors have an explanation for this?

Yes, this observation is correct. As mentioned in the text, the data are evaluated at relatively low cutoffs also because the selection did not show excellent agreement between codon substitutions resulting in the same amino acid change. We attribute this to the fact that the ATGL interactions in the Y2H system do not give very strong growth readouts (compared to non-related strong Y2H pairs), in other words because of the nature of the Y2H interactions, the selection was not super stringent. We were aware about this fact from

the beginning, however to solve the biological questions asked we needed to go with the Y2H PPIs and their respective growth strength in the rY2H setup.

2. The authors mention that before performing LUMIER they verified that all ATGL mutants were similarly expressed using Western Blot, however I don't see these blots presented anywhere. Please include this data.

We included the blots as Supplemental Figure 1.

3. I'm unclear on the data presented in Figure 3C. The authors mention they are looking at 'switch' mutations that only affect interaction with a single partner, however the data presented in Figure 3C suggests that some of these mutants still affect interaction reasonably well with more than one partner (e.g., I70A, P86K, L102K). The authors do touch on this briefly in the last paragraph of page 7, but I feel they are understating the level of effect. I think the authors should be more up front about describing these mutants as affecting certain interactions more strongly, as opposed to saying other interactions are 'unaffected'.

OK, we selected as switch variant those that affected one of the partners most selectively. We agree that not all switch mutations are of the same selectivity, in particular the PLIN5 variants are not as selective as others. This is however clearly shown in Figure 3C. We address this point more specifically on page 7 ("ideally ... unaffected"), and rephrased the text to be more up front on the selectivity of the switch variants on page 11 first paragraph.

[Reviewer #1 had no further remarks to the author]

Reviewer #2 (Remarks to the Author):

The authors have adequately addressed most of my initial concerns; however, a couple of points remain:

1. In response to points #2-4, the authors generated 293 cell lines with inducible expression of PLIN1 and PLIN5 and presented new imaging data in new Suppl. Fig2. Firstly, it was unclear why protein expression was only measured by mass spec. How specific and quantitative is this approach? Why not use a straightforward Western blot? Secondly, the confocal images shown in the new Suppl. Fig2 are of too low magnification and resolution to show distinct protein localization to the LD surface. To substantiate the conclusions, immunofluorescence images of higher magnification and quality are needed. The authors may consider treating cells with oleic acid to enlarge LDs.

2. Regarding point #7, is it possible to consider that the ATGL-CGI-58 interaction is not required for ATGL coactivation? In PMID: 26350461, previous work by the Oberer lab showed that a N-terminal tryptophan-rich sequence in CGI-58 is required for LD localization but not ATGL interaction.

Reviewer #3 (Remarks to the Author):

I have reviewed the experiments you conducted, addressing all the questions from my initial review. Thank you for your diligence.

Point by point reply:

Reviewer 2:

The authors have adequately addressed most of my initial concerns; however, a couple of points remain:

1. In response to points #2-4, the authors generated 293 cell lines with inducible expression of PLIN1 and PLIN5 and presented new imaging data in new Suppl. Fig2. Firstly, it was unclear why protein expression was only measured by mass spec. How specific and quantitative is this approach? Why not use a straightforward Western blot? Secondly, the confocal images shown in the new Suppl. Fig2 are of too low magnification and resolution to show distinct protein localization to the LD surface. To substantiate the conclusions, immunofluorescence images of higher magnification and quality are needed. The authors may consider treating cells with oleic acid to enlarge LDs.

First point: We present the mass spec data confirming PLIN1 or PLIN5 expression respectively, because these data are very reliable and quantitative. The PLIN expression is normalized to the expression of the whole proteome *i.e.* about 8600 quantified proteins in each measurement. These two proteins by far represent the largest changes found in the whole proteomes of the isogenic cell lines. In addition, we also monitor ATGL and CGI-58 expression in parallel (shown in the bar graph). However, we have done a western blot analysis as well and present the blots in an additional Figure panel in the revised Supplemental Figure 2. The western blot shows that the stable cell lines express PLIN1 and PLIN5 upon induction, respectively.

Second point:

In Supplemental Figure 2, cells were oleic acid treated for 6h and droplet size was larger in the cell lines that stably expressed PLIN1 or 5. The data were recorded on a state-of-the-art Leica SP5 confocal microscopy at 63x resolution. We included images presenting a set of representative cells (transfected and non-transfected) that clearly show the extend of co-localization of ATGL and LDs across cells. However, we agree that the LDs in the images were very small. We therefore include in the revised version images of single cells that allow for better visualization of single LDs. Some of the Plin-LDs are fairly large (μm -sized) and allow for visualization of ATGL and LDs in support of our conclusions.

In general, the LD size is below the diffraction limit (Abbé's limit) in both HeLa and HEK293 cells. Thus, it is not possible to tell whether ATGL is on a rim around the LDs or inside. Only colocalization of ATGL and LD within the resolution limit can be confirmed. With the high numerical aperture of the objective that we use (NA=1.4) and the confocal set-up, we have optimized the resolution as much as possible. To achieve resolution beyond these limits, super-resolution microscopy would be the necessary next step. We acknowledge that some of the published images in literature, which are also diffraction-limited, may suggest rim-structures. However, they are recorded in adipocytes or cell without any lipolysis activity so that LDs can get a size of a few μm . Our functional experiments required using cells with an active lipolysis machinery that, however, do not develop LDs of such large sizes.

In summary, we additionally provide a novel set of images with higher magnification, to a single cell size, and clearly show colocalization of ATGL and LDs

2. Regarding point #7, is it possible to consider that the ATGL-CGI-58 interaction is not required for ATGL coactivation? In PMID: 26350461, previous work by the Oberer lab showed that a N-terminal tryptophan-rich sequence in CGI-58 is required for LD localization but not ATGL interaction.

The paper by Oberer et al. showed that W 21, W25, and W29 of CGI-58 are jointly required for LD localization in cells and that in particular the W21A/W25A double mutant variant is more distributed in the cytoplasm and did not activate mATGL *in vitro* (no activity test in cells were reported). The interaction with ATGL was not tested. In fact the authors concluded: "When we tested a W21A/W29A variant, the ability of CGI-58 to localize to LDs and to activate ATGL was completely abrogated. ... Therefore, it is conceivable that the CGI-58 LD anchor undergoes a conformational change upon CGI-58 binding to ATGL. The LD anchor might be necessary for the correct orientation of CGI-58 on LDs, which provides the platform for interaction with ATGL, or for positioning TGs favorably with respect to CGI-58 bound ATGL."

The study therefore described variants on CGI-58, which render CGI-58 inactive *in vitro*. Our ATGL mutations on the other hand are active *in vitro* and stimulated by CGI-58, so they have distinct phenotypes. Oberer et al. do not assay lipolysis in cells. They clearly separate interaction from localization in their discussion. While we agree with the reviewer (and state this in the manuscript on page 12) that other mechanisms explaining the strong effect of the CGI-58 ATGL switch double mutant on lipolysis *in vivo* can be possible, the work of Oberer et al. (focusing only on CGI-58 rather than ATGL) neither supports our findings nor is in contradiction. Our work presents rigorous protein interaction measurements that guided *in vitro* and *in vivo* functional analyses. Therefore, the mechanistic interaction hypothesis is well supported by our data. However, we added the reference and mention that "Also CGI-58 mutations, such as W21A/W25A, that change LD anchoring of the activator, impaired activation of the enzymatic activity of ATGL." (page12, first paragraph)

Reviewer #2 (Remarks to the Author):

"In general, the LD size is below the diffraction limit (Abbé's limit) in both HeLa and HEK293 cells. Thus, it is not possible to tell whether ATGL is on a rim around the LDs or inside. Only colocalization of ATGL and LD within the resolution limit can be confirmed. With the high numerical aperture of the objective that we use (NA=1.4) and the confocal set-up, we have optimized the resolution as much as possible. To achieve resolution beyond these limits, super-resolution microscopy would be the necessary next step. We acknowledge that some of the published images in literature, which are also diffractionlimited, may suggest rim-structures. However, they are recorded in adipocytes or cell without any lipolysis activity so that LDs can get a size of a few μm . Our functional experiments required using cells with an active lipolysis machinery that, however, do not develop LDs of such large sizes."

Numerous papers have shown clear rim-like staining of LD proteins in nonadipocyte cells such as HeLa cells, which did not involve using a super resolution confocal. Please refer to early papers by Jackson lab on ATGL (PMID: 16239926; PMID: 21789191) and many papers on LDs by Toby Walther group.